# Summer warming explains widespread but not uniform greening in the Arctic tundra biome

Logan T. Berner [1✉], Richard Massey [1], Patrick Jantz [1], Bruce C. Forbes [2], Marc Macias-Fauria [3], Isla Myers-Smith [4], Timo Kumpula[5], Gilles Gauthier [6], Laia Andreu-Hayles [7], Benjamin V. Gaglioti[8], Patrick Burns [1], Pentti Zetterberg[9], Rosanne D'Arrigo[7] & Scott J. Goetz [1]

Arctic warming can influence tundra ecosystem function with consequences for climate feedbacks, wildlife and human communities. Yet ecological change across the Arctic tundra biome remains poorly quantified due to field measurement limitations and reliance on coarse-resolution satellite data. Here, we assess decadal changes in Arctic tundra greenness using time series from the 30 m resolution Landsat satellites. From 1985 to 2016 tundra greenness increased (greening) at ~37.3% of sampling sites and decreased (browning) at ~4.7% of sampling sites. Greening occurred most often at warm sampling sites with increased summer air temperature, soil temperature, and soil moisture, while browning occurred most often at cold sampling sites that cooled and dried. Tundra greenness was positively correlated with graminoid, shrub, and ecosystem productivity measured at field sites. Our results support the hypothesis that summer warming stimulated plant productivity across much, but not all, of the Arctic tundra biome during recent decades.

[1] School of Informatics, Computing, and Cyber Systems, Northern Arizona University, Flagstaff, AZ 86011, USA. [2] Arctic Centre, University of Lapland, 96101 Rovaniemi, Finland. [3] School of Geography and the Environment, University of Oxford, Oxford OX1 3QF, UK. [4] School of GeoSciences, University of Edinburgh, Edinburgh EH9 3FF, UK. [5] Department of Geographical and Historical Studies, University of Eastern Finland, 80101 Joensuu, Finland. [6] Department of Biology and Centre d'études nordiques, Université Laval, Quebec City, QC G1V0A6, Canada. [7] Lamont-Doherty Earth Observatory, Columbia University, Palisades, NY 10964, USA. [8] Water and Environment Research Center, University of Alaska Fairbanks, Fairbanks, AK 99775, USA. [9] Department of Forest Sciences, University of Eastern Finland, 80101 Joensuu, Finland. ✉email: logan.berner@nau.edu

The Arctic tundra biome is rapidly warming[1] with fundamental consequences for climate feedbacks[2], wildlife[3], and human communities[4]. Nevertheless, assessing the impacts of climate change on tundra ecosystems throughout the Arctic remains a significant challenge, as recently highlighted by the U.S. National Academy of Sciences[5]. Multi-decadal field measurements provide the most direct evidence of tundra response to warming, but such studies are scarce across the Arctic, especially in the Canadian and Eurasian Arctic[6]. Long-term field studies that do exist document recent increases in plant cover, growth, height, and biomass, and a shift towards shrub dominance in some tundra ecosystems[6–9], while other areas show little change in vegetation[10,11], or even warming-induced declines in plant growth[12,13]. Diverse ecological responses to warming and the paucity of long-term field measurements underscore the need for effectively using Earth-observing satellites to assess ecological changes that are occurring across one of Earth's coldest but most rapidly warming biomes.

Earth-observing satellites have been used to infer changes in tundra greenness since the 1980s, but pan-Arctic assessments historically relied on coarse spatial resolution satellite data sets that exhibit notable discrepancies through time. The Normalized Difference Vegetation Index (NDVI) provides a metric of tundra greenness that can be derived from satellite observations and is broadly related to tundra plant productivity[14] and aboveground biomass[15,16]. Pan-Arctic changes in NDVI since the 1980s have been exclusively assessed with the Advanced Very High-Resolution Radiometers (AVHRR)[17]. These satellites show increasing NDVI (greening) across large parts of the Arctic, but decreasing NDVI (browning) in several regions (e.g., Canadian High Arctic). However, the prevalence and spatial patterns of greening and browning differ considerably among AVHRR NDVI data sets[18]. These discrepancies partially reflect challenges with cross-calibrating sensors flown on 16 separate satellites[18,19]. Furthermore, the coarse spatial resolution of AVHRR NDVI data sets (typically ~8 km) far exceeds the scale of ecological change in heterogeneous tundra landscapes[10] and limits the ability to attribute recent trends to potential landscape level drivers (e.g., permafrost thaw, wildfires). Moreover, the coarse spatial resolution makes it difficult to reconcile trends with field observations[11]. These issues require caution in analyses based on the AVHRR satellites for pan-Arctic assessment of tundra response to warming and underscore the need for assessments using higher-resolution satellite observations that also extend back to the 1980s.

The high-resolution Landsat satellites offer a promising complement to the AVHRR satellites for assessing pan-Arctic trends in tundra greenness and identifying factors that have driven these changes. The Landsat satellites cover the same period as AVHRR but with fewer satellites, which reduces but does not eliminate challenges with cross-sensor calibration[20]. Furthermore, the Landsat satellites provide 30 m resolution observations that more closely match the scale of field measurements and ecological change than AVHRR observations. However, higher spatial resolution means that each location is observed fewer times each growing season, which contributes to the challenge of assessing vegetation phenology, especially since the growing season is often short and cloudy in the Arctic[21]. The high spatial resolution also increases data volume and thus the Landsat satellites have typically been used for local assessments of tundra greenness[22–24], although recent advances in computing[25] and remote sensing[26] have enabled regional to continental assessments in the North American Arctic[20,27]. Nevertheless, pan-Arctic changes in tundra greenness and their relation to climate, permafrost, and fire have not been assessed using the Landsat satellite series.

Here, we advance current understanding of recent changes in tundra greenness across the Arctic tundra biome (Fig. 1) using more than three decades of high-resolution Landsat satellite imagery in concert with a broad suite of environmental and field data sets. Specifically, we ask:

(1) To what extent did tundra greenness change during recent decades in the Arctic?
(2) How closely did inter-annual variation in tundra greenness track summer temperatures?
(3) Were tundra greenness trends linked with climate, permafrost, topography, and/or fire?
(4) How closely did satellite observations of tundra greenness relate to temporal and spatial variation in plant productivity measured at field sites?

To characterize tundra greenness, we use the annual maximum summer NDVI ($NDVI_{max}$) derived from surface reflectance measured by Landsat 5, 7, and 8. We first extract all available summer Landsat data for 50,000 random sampling sites in the Arctic using Google Earth Engine[25] and then generate annual time series of $NDVI_{max}$ for each vegetated sampling site. We develop novel approaches to not only further cross-calibrate NDVI among Landsat sensors, but also minimize biases associated with estimating annual $NDVI_{max}$ when few summer measurements were available. Estimates of annual $NDVI_{max}$ are sensitive to sensor radiometric calibration, cross-sensor calibration, and modeling approach, thus we propagate these sources of error and uncertainty in climate and field data sets through the analysis using Monte Carlo simulations ($n = 10^3$). After estimating annual $NDVI_{max}$ at each sampling site, we then assess changes in tundra greenness and covariation with summer temperatures from 1985 to 2016 and 2000 to 2016 using rank-based trend tests and correlations in a Monte Carlo uncertainty framework. Moreover, we examine the extent to which tundra

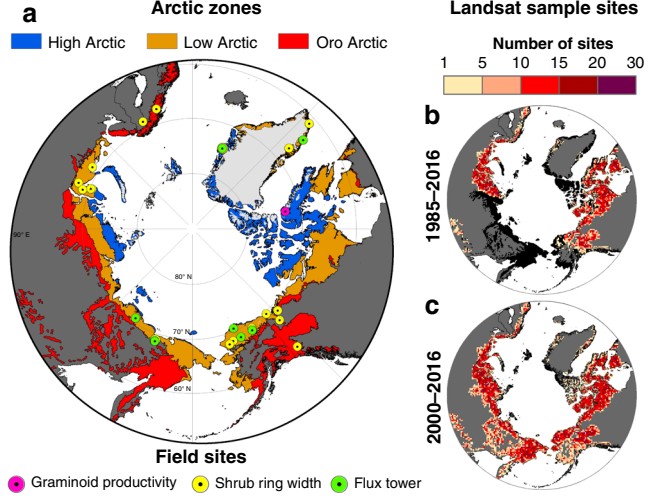

**Fig. 1 Spatial extent of Arctic tundra and locations of field and Landsat sample sites. a** The Arctic can be subdivided into the minimally vegetated High Arctic, moderately vegetated Low Arctic, and southern mountainous Oro Arctic. Landsat $NDVI_{max}$ was compared against three metrics of plant productivity measured at field sites around the Arctic. **b, c** Number of Landsat sampling sites within a $50 \times 50$ km$^2$ grid cell that were used for assessing $NDVI_{max}$ trends and correlations with summer temperatures from 1985 to 2016 and 2000 to 2016. It was not possible to assess $NDVI_{max}$ trends or correlations in the eastern Eurasian Arctic from 1985 to 2016 owing to the lack of Landsat data prior to circa 2000. Arctic tundra without adequate data for Landsat assessment is shown in black. Projection: Lambert Azimuthal Equal Area.

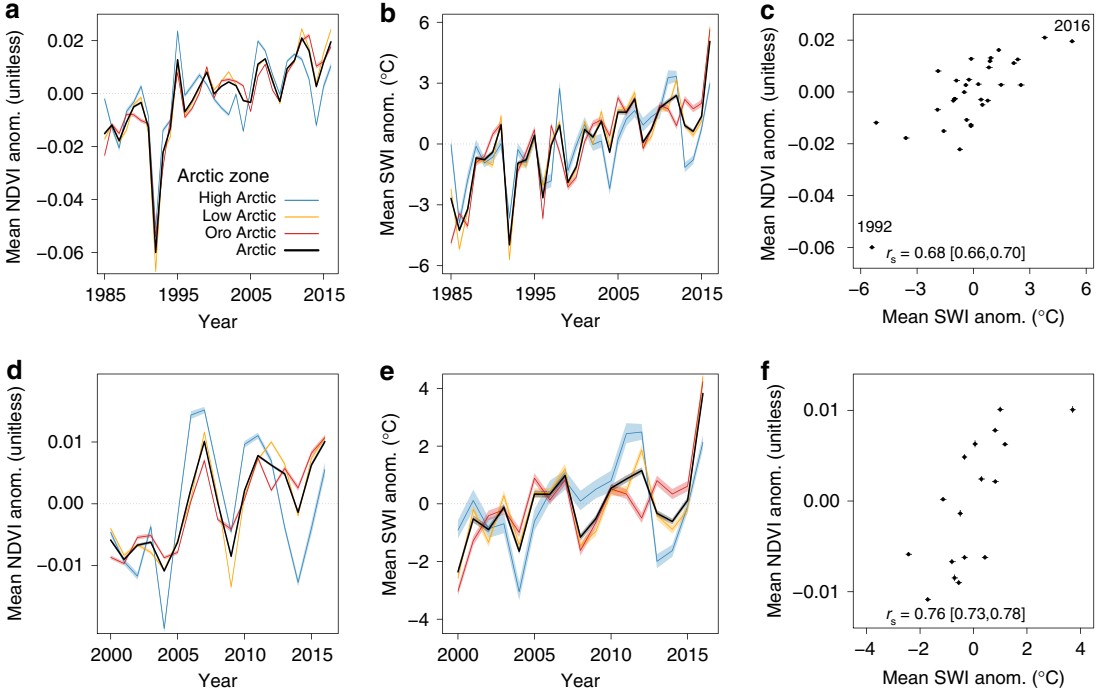

**Fig. 2 Tundra greenness and summer air temperature time series and covariation.** Left panels show changes in mean Landsat NDVI$_{max}$ [unitless] anomalies for the Arctic and each zone from 1985 to 2016 (**a**) and 2000 to 2016 (**d**). Middle panels show changes in mean summer warmth index [SWI; °C] anomalies from 1985 to 2016 (**b**) and 2000 to 2016 (**e**) derived from five temperature data sets. Right panels show the relationship between mean Arctic NDVI$_{max}$ and SWI anomalies from 1985 to 2016 (**c**) and 2000 to 2016 (**f**). Spearman's correlation coefficients ($r_s$) relating NDVI$_{max}$ and SWI are provided for each period. Narrow error bands and whiskers depict 95% confidence intervals derived from Monte Carlo simulations ($n = 10^3$). Note that while mean SWI time series are based on pan-Arctic data, the NDVI$_{max}$ time series, and NDVI$_{max}$–SWI relationships are based on sites where Landsat data were available from 1985 to 2016 (**a**, **c**) and 2000 to 2016 (**d**, **f**), as shown in Fig. 1.

greenness trends from 2000 to 2016 were linked with summer temperature, soil moisture, permafrost, topography, land cover, and fire using geospatial data sets and machine learning. Last, we compare Landsat observations of tundra greenness against three metrics of plant productivity at field sites across the Arctic. It is important to note that Landsat observations sufficient for time series analysis were available for ~64% and ~96% of the Arctic domain from 1985 to 2016 and 2000 to 2016, respectively, with particularly improved coverage across the eastern Eurasian Arctic during the more recent period (Fig. 1b, c). Our analysis reveals extensive but not uniform greening in the Arctic tundra biome during recent decades that tended to occur in warm areas with increasing summer air temperature, soil temperature, and soil moisture. Our findings are consistent with the hypothesis that summer warming stimulated plant productivity across much of the Arctic tundra biome during recent decades, which has consequences for climate feedbacks, wildlife, and human communities.

## Results

**Greening and warming of the Arctic tundra biome.** Our analysis of Landsat NDVI$_{max}$ and climatic data showed strong increases in average tundra greenness and summer air temperatures during the past three decades in the Arctic and constituent Arctic zones (Fig. 2 and Supplementary Table 2). Mean Arctic NDVI$_{max}$ increased 7.3 [7.0, 7.7]% from 1985 to 2016 and 3.6 [3.4, 3.7]% from 2000 to 2016 [95% Monte Carlo confidence intervals]. Changes in mean NDVI$_{max}$ from 1985 to 2016 were considerably higher in the Low Arctic and Oro Arctic than the High Arctic; however, the High Arctic experienced the highest percent increase in mean NDVI$_{max}$

from 2000 to 2016 (Supplementary Table 2). These positive trends in mean NDVI$_{max}$ indicate systematic greening of the Arctic tundra biome during the past three decades.

Greening of the Arctic occurred in concert with a rapid increase in summer air temperatures over the past three decades. We quantified summer temperatures with the summer warmth index (SWI) computed as the annual sum of mean monthly air temperatures >0 °C (units: °C)[28] using an ensemble of five temperature data sets. The mean Arctic SWI increased 5.0 [4.9, 5.1] °C from 1985 to 2016 and 2.5 [2.3, 2.7] °C from 2000 to 2016, with warming evident in each Arctic zone (Fig. 2b, e and Supplementary Table 6). Annual mean Arctic NDVI$_{max}$ and SWI anomalies were positively correlated from 1985 to 2016 (Spearman's correlation [$r_s$] = 0.68 [0.66, 0.70]) and 2000 to 2016 ($r_s$ = 0.76 [0.73, 0.78]; Fig. 2c, f), particularly when SWI was averaged over the current and preceding year ($r_s$ = 0.86 [0.85, 0.88] and 0.89 [0.88, 0.91], respectively). Correlations were weaker, but still positive, when each time series was linearly detrended (Supplementary Table 7). For instance, the correlations between annual NDVI$_{max}$ and SWI decreased to 0.43 [0.41, 0.47] and 0.39 [0.33, 0.46] for the periods 1985 to 2016 and 2000 to 2016, respectively. Positive NDVI$_{max}$–SWI correlations were also evident in each Arctic zone (Supplementary Table 7). The lowest mean Arctic NDVI$_{max}$ occurred in 1992 following acute cooling caused by the massive eruption of Mount Pinatubo[29], whereas the highest mean Arctic NDVI$_{max}$ occurred during record-setting warm summers in 2012 and 2016. Strong positive NDVI$_{max}$ trends and NDVI$_{max}$–SWI correlations during a period of rapid warming suggest that reductions in temperature limitations on biological and/or biogeochemical processes could have contributed to recent increases in tundra greenness in the Arctic tundra biome.

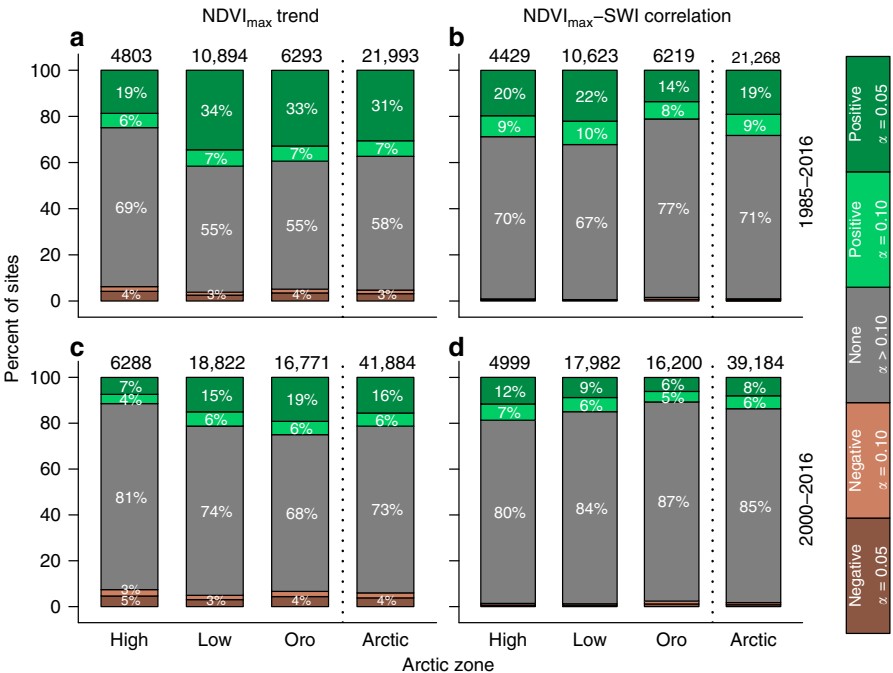

**Fig. 3 Tundra greenness trends and correlations with summer air temperatures summarized by Arctic zone.** Left panels show the percent of sites in the Arctic and each zone that exhibited a positive trend [green], a negative trend [brown], or no trend [gray] in Landsat NDVI$_{max}$ from 1985 to 2016 (**a**) and 2000 to 2016 (**c**). Right panels show the percent of sites that exhibited a positive correlation [green], a negative correlation [brown], or no correlation [gray] between annual NDVI$_{max}$ and the summer warming index (SWI) from 1985 to 2016 (**b**) and 2000 to 2016 (**d**). In all panels, dark and light shades denote significance levels of trends or correlations [dark shades: $\alpha = 0.05$; light shades: $\alpha = 0.10$). The sample size is provided above each bar.

**Spatial variability and drivers of tundra greenness trends**. Our biome-scale analysis indicated an overall greening of the Arctic tundra biome that closely corresponded with summer warming in recent decades; however, tundra greenness was stable or even declined in some areas and inter-annual variability in tundra greenness was often weakly related to summer temperatures at individual sampling sites. Landsat NDVI$_{max}$ increased (critical value [$\alpha$] = 0.10; greening) at 37.3 [36.3, 38.4]% of sampling sites [95% Monte Carlo confidence interval] and decreased ($\alpha = 0.10$; browning) at 4.7 [4.4, 5.2]% of sites from 1985 to 2016, although exhibited no trend at 58.0 [57.1, 58.7]% of sampling sites (Fig. 3a and Supplementary Table 3). Similarly, greening occurred at 21.3 [20.8, 21.7]% of sampling sites and browning at 6.0 [5.8, 6.3]% of sampling sites from 2000 to 2016 (Fig. 3c). Greening was thus 7.9 [7.1, 8.7] and 3.6 [3.4, 3.8] times more common than browning during these two periods and occurred at a higher percentage of sites in the Oro Arctic and Low Arctic than in the High Arctic (Fig. 3a, c). There was extensive greening in parts of western Eurasia (e.g., Gydan Peninsula and southern Yamal Peninsula) and North America (e.g., Ungava Peninsula, Northwest Territories, and northwestern Nunavut) from 1985 to 2016. The increase in availability of observations from 2000 to 2016 also revealed extensive greening in eastern Eurasia (e.g., Chukotka and mountains of Yakutia; Fig. 4a, e). Although browning was much less common than greening, it was evident at sampling sites widely distributed across the domain and occurred at a slightly higher percentage of sites in the High Arctic and Oro Arctic than the Low Arctic (Figs. 3 and 4). Annual NDVI$_{max}$ and SWI were positively correlated ($\alpha = 0.10$) at 28.2 [27.3, 29.1]% of sampling sites ($r_s = 0.41 \pm 0.06$; mean $\pm$ 1 SD) and negatively correlated ($\alpha = 0.10$) at 1.0 [0.8, 1.1]% of sampling sites ($r_s = -0.40 \pm 0.06$; Fig. 3b) from 1985 to 2016, with positive NDVI$_{max}$–SWI correlations at 41.0 [39.5, 42.5]% of sampling sites that greened and negative correlations at 6.5 [5.0, 8.0]% of sampling sites that

browned. There was a lower frequency of significant correlations between annual NDVI$_{max}$ and SWI from 2000 to 2016. Overall, greening was prevalent and often associated with summer temperatures in the Oro Arctic and Low Arctic, while browning was uncommon, but widely distributed.

To further explore potential drivers of changes in tundra greenness among sampling sites, we constructed Random Forest models to associate the NDVI$_{max}$ trend category from 2000 to 2016 (i.e., browning, no trend, or greening) with climate, permafrost, land cover, topography, and fire (Supplementary Table 8). Cross-validated model classification accuracy was 55 [53, 58]%, but the classification accuracies for greening and browning classes were 70 [68, 73]% and 73 [70, 75]%, respectively (Supplementary Tables 9 and 10). The expected classification accuracy at random would be 33.3%. The six most important predictor variables included change in SWI (2000–2016), annual mean soil temperature (1 m depth), and SWI in the early 2000s, elevation, change in minimum summer soil moisture (2000–2016), and change in annual mean soil temperature (2003–2016; Fig. 5a). Greening occurred more often at warm, high-elevation sampling sites with increased summer air temperatures, annual mean soil temperatures, and summer soil moisture. Conversely, browning occurred more often at cold, low-elevation sampling sites with decreased summer air temperatures, annual mean soil temperatures, and summer soil moisture (Fig. 5b). A notable exception was the sharp decline in greening and increase in browning where soil temperatures in the early 2000s exceeded 0 °C. It is also notable that at a pan-Arctic scale recent fires were not an important predictor of greening or browning, reflecting the fact that fires occurred at only ~1.1 % of sampling sites from 2001 to 2016.

**Covariation of tundra greenness and plant productivity.** To validate interpretation of recent greening and browning trends, we compared Landsat NDVI$_{max}$ against several metrics of spatial

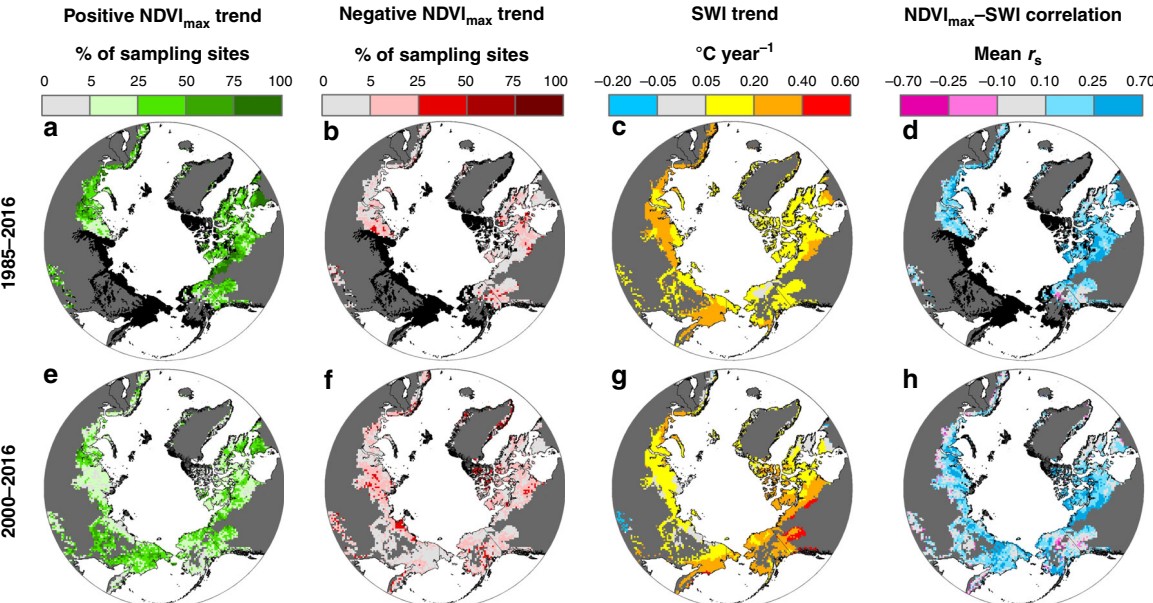

**Fig. 4 Tundra greenness and summer air temperature trends and correlations across the Arctic.** Top panels (**a–d**) depict Landsat $NDVI_{max}$ trends, summer warmth index (SWI) trends, and $NDVI_{max}$–SWI correlations from 1985 to 2016, while bottom panels (**e–h**) depict trends and correlations from 2000 to 2016. Trends in tundra greenness were inferred at random sampling sites (Fig. 1b, c) using $NDVI_{max}$ time series and Mann–Kendall trend tests. The percent of sites with positive (**a**, **e**) and negative (**b**, **f**) trends [$\alpha = 0.10$] was summarized within $50 \times 50$ km$^2$ grid cells. **c**, **g** Changes in annual SWI derived using an ensemble of five temperature data sets. **d**, **h** Mean Spearman's correlation ($r_s$) between annual $NDVI_{max}$ and SWI among sites within each $50 \times 50$ km$^2$ grid cell. The maps also depict areas in the Arctic that are barren [mean $NDVI_{max} < 0.10$; dark gray] or lacked adequate Landsat data for trend and correlation assessments [black].

and temporal variability in plant productivity at field sites around the Arctic (Fig. 1 and Supplementary Fig. 7). The plant productivity metrics included graminoid aboveground net primary productivity (ANPP; g dry matter m$^{-2}$ year$^{-1}$) estimated from clip harvests, shrub ring-width indices (RWIs; unitless) derived from measurements of annual stem radial growth, and ecosystem gross primary productivity (GPP; g C m$^{-2}$ year$^{-1}$) estimated from measurements by eddy covariance flux towers. We found annual landscape median $NDVI_{max}$ and graminoid ANPP were positively correlated from 1990 to 2017 at a long-term monitoring site on Bylot Island in far northern Canada ($r_s = 0.43$ [0.24, 0.58]; Supplementary Fig. 8). Moreover, the $NDVI_{max}$–ANPP relationship was stronger when $NDVI_{max}$ was averaged over the two preceding years ($r_s = 0.68$ [0.55, 0.78]; Supplementary Fig. 8). We also examined the temporal correspondence between annual detrended $NDVI_{max}$ and 22 shrub RWI chronologies representing alder (*Alnus* spp.), willow (*Salix* spp.), and dwarf birch (*Betula* spp.) in six Arctic countries. The $NDVI_{max}$–shrub RWI correlations ($r_s$) ranged from $-0.12$ [$-0.33$, 0.04] to 0.84 [0.72, 0.93] with a median $r_s$ of 0.42 [0.34, 0.50] among chronologies (Supplementary Fig. 9 and Supplementary Table 11). Last, we found a positive correlation between spatial patterns of median annual $NDVI_{max}$ and GPP across 11 flux tower sites that were part of the Arctic Observing Network or FLUXNET ($r_s = 0.72$ [0.54, 0.88]; Supplementary Fig. 10). This suite of comparisons shows Landsat $NDVI_{max}$ positively corresponds with metrics of graminoid, shrub, and ecosystem productivity either through time or across tundra ecosystems.

## Discussion

We provide here a pan-Arctic assessment of changes in tundra greenness using high-resolution Landsat $NDVI_{max}$ and evaluate links between tundra greenness and field measurement of plant productivity. We found widespread greening in recent decades that was linked with increasing summer air temperatures, annual

soil temperatures, and summer soil moisture; however, tundra greenness had no significant trend in many areas and even declined in others. Our assessment relied on carefully cross-calibrated and phenologically modeled estimates of $NDVI_{max}$ that we show were positively correlated with temporal and spatial variability in tundra plant productivity (i.e., graminoid ANPP, shrub radial growth, and ecosystem GPP; see Supplementary Discussion). Prior regional studies related positive trends in Landsat NDVI with increasing tundra shrub cover[23,30] and spatial variability in Landsat NDVI with tundra plant aboveground biovolume[31] and biomass[16]. Consequently, we interpret the observed tundra greening as evidence that plant productivity, height, biomass, and potentially shrub dominance increased since the 1980s in large parts of the Arctic in response to recent summer warming. This interpretation of tundra greening is broadly supported by changes in vegetation observed by Inuit communities in northeastern Canada[32] and Nenet herders in northwestern Russia[33], as well as documented by long-term field surveys[6–8], dendroecology[10,34], and high-resolution remote sensing[35,36] at sites around the Arctic. Nevertheless, attribution of local tundra greening to specific biological changes and environmental drivers remains an important challenge that will require further mapping and modeling of potential drivers at higher spatial and thematic resolution, coupled with field measurements, across the Arctic.

While our Landsat analysis revealed an increase in mean Arctic $NDVI_{max}$ that broadly supports biome-scale changes inferred using coarser-resolution AVHRR (1982 onward at ~8 km resolution) and MODIS (2000 onward at 500 m resolution) data sets, it also highlights inconsistencies among satellites[17,18,37]. For instance, we report a 7.9 [7.1, 8.7]:1 ratio of greening to browning from 1985 to 2016. On the other hand, recent analyses of AVHRR (GIMMS$_{3g}$) data suggested ratios of 29:1 and 14:1 from 1982 to 2008[18] and 1982 to 2014[38], respectively, while analysis of MODIS data suggested a ratio of 13:1 from 2000 to 2018[37]. Different time

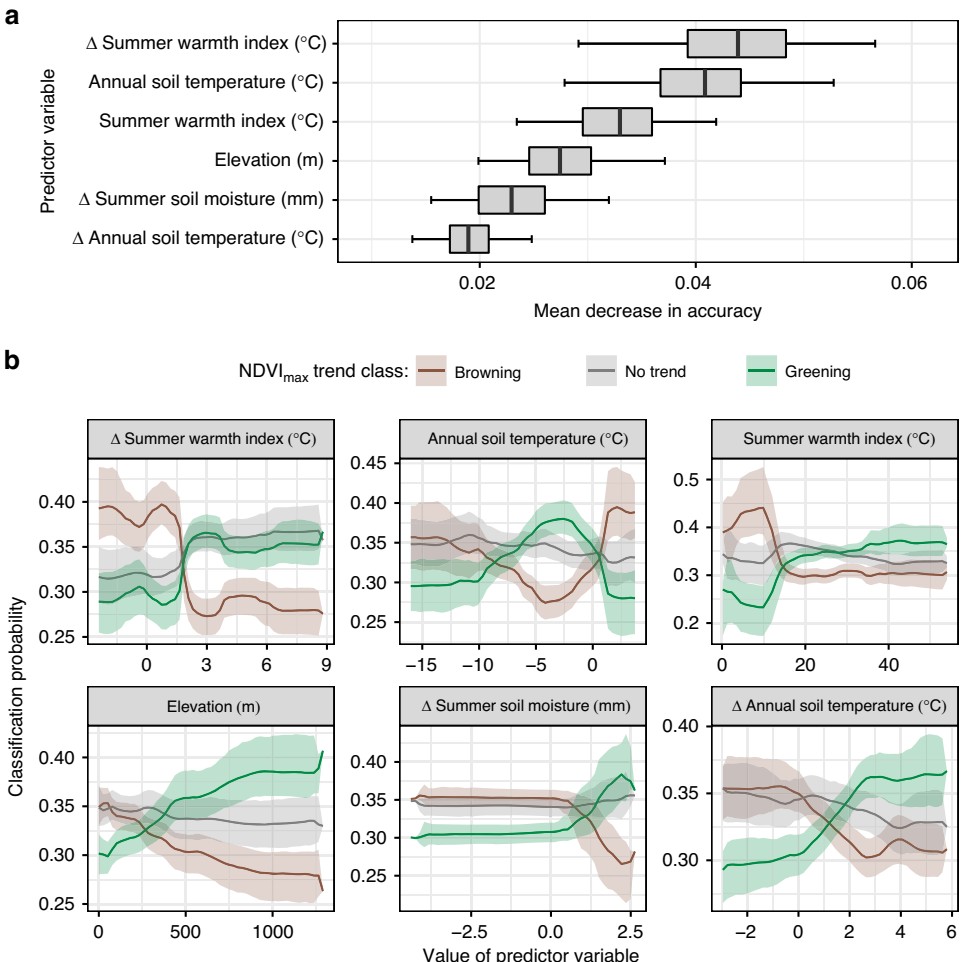

**Fig. 5 Primary environmental predictors of tundra greenness trends.** Variable importance and partial dependence of the six most important variables for predicting site-level Landsat NDVI$_{max}$ trend categories from 2000 to 2016 using Random Forests. The three NDVI$_{max}$ trend categories included browning, no trend, and greening that were based on trend direction and significance ($\alpha = 0.10$). **a** Variable importance was characterized by the mean decrease in accuracy, where a higher value indicates that a variable is more important to the classification. **b** Partial dependency plots illustrate how each predictor variable affects class probability while accounting for the mean effect of other predictors in the model. The top six predictor variables included changes ($\Delta$) in summer warmth index (2000–2016), summer soil moisture (2000–2016), and annual mean soil temperature (2003–2016), as well as elevation, summer warmth index in 2000, and annual mean soil temperature in 2003. Soil temperature data were for 1 m depth and were not available prior to 2003. Boxplot lines and whiskers in **a** depict median estimates and 95% confidence intervals derived from Monte Carlo simulations ($n = 10^3$), as do solid lines and error bands in **b**.

periods, spatial domains, and NDVI metrics hinder direct comparison among these studies, but these initial comparisons suggest that Landsat either detects less greening and/or more browning relative to AVHRR (GIMMS$_{3g}$) and MODIS.

Compared to AVHRR or MODIS, high-resolution Landsat observations may be especially advantageous when assessing vegetation dynamics in heterogeneous tundra landscapes[37] where micro- and macro-topography shape plant communities that are often interspersed with surface water, snow, and barren ground[16,37,39]. Patches of greening and browning can occur in close proximity on a landscape, leading coarse-resolution observations to integrate divergent trajectories of change[20,37]. Consequently, local browning could be obscured by widespread stability or greening on a landscape when viewed at coarse resolution. Similarly, a predominantly stable landscape could be perceived as greening at coarse resolution given enough hotspots of change. The spatial scale of Landsat imagery is much closer to the relevant scales of ecological processes than AVHRR or MODIS and thus potentially captures ecological stability and change in a more realistic way. However, the utility of Landsat is constrained by the

lack of observations in parts of the Arctic (e.g., eastern Eurasia) prior to 2000 and by the lower frequency of observations acquired each summer, which we partially addressed by modeling annual NDVI$_{max}$ with site-specific information on land surface phenology. Newer satellites (e.g., Sentinel-2, Planet, Worldview-3) and unmanned aerial vehicles[40] can facilitate higher-resolution mapping of tundra properties and could be combined with decades of Landsat observations to better understand recent changes in Arctic tundra.

Regional consistencies and inconsistencies were evident among Landsat, AVHRR, and MODIS satellite time series. For example, these satellite time series consistently show extensive greening in the eastern Eurasian Arctic, providing evidence of recent ecological change across a large region underlain by continuous permafrost that has little to no long-term, ground-based ecosystem monitoring[6,41]. On the other hand, our study and prior regional Landsat assessments[20,22] show pronounced greening in northern Quebec, where AVHRR and MODIS suggest modest greening. Greening in this region is likely associated with observed increases in graminoid and shrub cover, particularly of dwarf

birch (*Betula glandulosa*)[22,36]. Our Landsat analysis also indicated recent browning along the rugged southwestern coast of Greenland that is consistent with local declines in shrub growth[12], but not evident in assessments that used AVHRR or MODIS. The link between recent browning and declining shrub growth is further supported by the positive correlations that we found between annual Landsat $NDVI_{max}$ and stem radial growth of grayleaf willow (*Salix glauca*; $r_s = 0.60$ [0.39, 0.78]) and dwarf birch (*Betula nana*; $r_s = 0.61$ [0.45, 0.74]) in this region (Supplementary Table 12). Overall, the Landsat, AVHRR, and MODIS satellites show extensive greening and modest browning in the Arctic tundra biome during recent decades; however, regional discrepancies in greening and browning highlight the need for rigorous comparisons among satellites and between satellite and field measurements.

We found no trend in tundra greenness at most locations, despite pervasive increases in summer air temperatures. It is possible that indirect drivers of vegetation change, such as permafrost thaw and nutrient release, are accumulating in response to warming of summer air temperatures, or that plants are limited by other environmental constrains. Low soil temperatures, nutrients, and moisture can limit plant response to rising air temperatures[10,42], as can strong genetic adaptation to prevailing environmental conditions[43]. In other cases, warming might have stimulated plant growth, but led to no change in tundra greenness due to grazing, browsing, and trampling by herbivores. Field[44] and modeling[45] studies show that herbivory can significantly suppress tundra response to warming, although effects of vertebrate and invertebrate herbivores on Arctic greening and browning remain unclear. Last, tundra greenness could, in some areas, be confounded by patchy vegetation being interspersed with bare ground, surface water, or snow[24,37]. Despite limitations with NDVI (e.g., see ref. [37] for recent tundra-specific review), our results indicate Arctic plants did not universally benefit from warming in recent decades, highlighting diverse plant community responses to warming likely mediated by a combination of biotic and abiotic factors.

Our analysis showed that tundra browning occurred at a small percentage (~5%) of sampling sites during recent decades, and although uncommon, it was widely distributed in the Arctic. Inter-annual variability in tundra greenness and summer air temperatures were negatively correlated at only ~6% of sites that browned, suggesting little direct link between warming and browning on an annual time scale. Our Landsat analysis did detect browning in western Greenland from 2000 to 2016 that aligns with field studies showing recent declines in deciduous shrub growth due to warming-induced drought stress[12,13], as well as defoliation from moths (*Eurois occulta*) and increased browsing by muskoxen (*Ovibos moschatus*)[12]. On the other hand, we found that browning from 2000 to 2016 was most probable at sampling sites where summer air temperature, annual soil temperature, or summer soil moisture decreased; however, cooling and drying infrequently occurred in the Arctic during the 2000s. Cooler and drier conditions could suppress tundra plant growth and have contributed, for instance, to recent tundra browning detected by Landsat in parts of the Yablonovy Mountains near Lake Baikal in central Siberia. Other potential drivers of browning in the Arctic include local changes in surface hydrology (wetting or drying) associated with permafrost degradation[46–48], extreme weather events[49,50], and industrial development[24,51]. In concurrence with terrestrial biosphere models[29,52], our analysis suggests that warming tended to promote rather than suppress plant productivity and biomass in the Arctic during recent decades, but increasing frequency of permafrost degradation, extreme weather events, pest outbreaks, and industrial development could contribute to future browning[49].

Tundra fires are another contributor to greening and browning in the Arctic[53]; however, our results indicate that their contribution is currently quite small at a pan-Arctic extent due to their infrequent occurrence. Examining MODIS satellite observations[54] from 2001 to 2016, we found that 1.1% of sampling sites burned over the 16 years period, which suggests a current fire rotation of ~1450 years for the Arctic tundra biome. Regional fire rotation within the biome is strongly governed by summer climate and is considerably shorter (~425 years) in the warmest and driest tundra regions (e.g., Noatak and Seward, Alaska)[53,55]. Our analysis further showed that fires recently occurred at ~1.0% of sampling sites that greened and ~2.4% of sampling sites that browned. Tundra fires can emit large amounts of carbon into the atmosphere[56] and lead to temporary browning by burning off green plants, while subsequent increases in soil temperature and permafrost active layer depth can stimulate a long-term increase in plant growth and shrub dominance in some but not all cases[53,57,58]. Continued warming will likely increase annual area burned in the tundra biome[55]; thus, fires could become a more important driver of tundra greening and browning in the Arctic over the twenty-first century.

Our analysis contributes to a growing body of evidence showing recent widespread changes in the Arctic environment that can impact climate feedbacks. Rising temperatures are likely stimulating carbon uptake and storage by plants in areas that are greening (negative climate feedback), but also leading to soil carbon loss by thawing permafrost and enhancing microbial decomposition (positive climate feedback)[59,60]. Moreover, greening can reduce surface albedo as plants grow taller and leafier (positive climate feedback)[61,62] while also affecting soil carbon release from permafrost thaw by altering canopy shading and snow-trapping (mixed climate feedbacks)[63]. The net climate feedback of these processes is currently uncertain; thus, our findings underscore the importance of future assessments with Earth system models that couple simulations of permafrost, vegetation, and atmospheric dynamics at moderately high spatial resolution.

Widespread tundra greening can also affect habitat suitability for wildlife and semi-domesticated reindeer, with consequences for northern subsistence and pastoral communities. As an example, moose[64] and beavers[65] recently colonized, or recolonized, increasingly shrubby riparian habitats in tundra ecosystems of northern Alaska and thus appear to be benefiting from recent tundra greening. Conversely, caribou populations in the North American Arctic could be adversely affected if warming stimulates vascular plant growth at the expense of lichens, an important winter forage[66]. In the western Eurasian Arctic, indigenous herders (e.g., Sami, Nenets) manage about two million semi-domesticated reindeer on tundra rangelands. Shrub growth, height, and biomass significantly increased on these rangelands in recent decades, while lichen cover and biomass declined mostly due to trampling during the snow-free period[34,51,67]. Our Landsat analysis showed tundra greening in regions with potential moose, beaver, caribou, and reindeer habitat and demonstrated that variability in tundra greenness was often associated with annual shrub growth in these regions (Supplementary Table 12). Many northern communities rely on subsistence hunting or herding and thus changes in wildlife or herd populations can influence food security[4,68] and dietary exposure to environmental contaminants[69]. By documenting the extent of recent greening, analyses such as ours can help identify where wildlife and northern communities might be most impacted by ongoing changes in vegetation.

In summary, we assessed pan-Arctic changes in tundra greenness using high spatial resolution Landsat satellite observations and found evidence to support the hypothesis that recent

summer warming contributed to increasing plant productivity and biomass across substantial portions of the Arctic tundra biome during the past three decades. Nevertheless, we also document summer warming in many areas that did not become greener. The lack of greening in these areas points towards lags in vegetation response and/or to the importance of other factors in mediating ecosystem response to warming. Sustained warming may not drive persistent greening in the Arctic over the twenty-first century for several reasons, particularly hydrological changes associated with permafrost thaw, drought, and fire. Overall, our high spatial resolution pan-Arctic assessment highlights tundra greening as a bellwether of global climatic change that has wide-ranging consequences for life in northern high-latitude ecosystems and beyond.

## Methods

**Generating annual Landsat NDVI$_{max}$ time series**. We developed annual estimates of maximum summer NDVI[70] from 1985 to 2016 (NDVI$_{max}$) for 50,000 sampling sites in Arctic tundra[71] using 30 m resolution measurements of surface reflectance from the Landsat satellites (Landsat Collection 1)[72,73]. We first buffered each site by 50 m (radius) and then used the Google Earth Engine[25] Python[74] interface to extract all Landsat 5, 7, and 8 surface reflectance measurements acquired June through August from 1984 to 2016. This yielded 507 million multi-band surface reflectance measurements, of which 112 million (28%) were considered to be useable clear-sky measurements based on the CFmask algorithm[26,75] and scene criteria. We then used these clear-sky measurements to estimate annual NDVI$_{max}$ while considering multiple sources of uncertainty. Estimates of annual NDVI$_{max}$ are sensitive to radiometric calibration uncertainty and systematic differences in NDVI among Landsat sensors, as well as to the availability and seasonal timing of measurements. We therefore developed new techniques to further cross-calibrate NDVI among sensors and model annual NDVI$_{max}$. Furthermore, we characterized uncertainty in our estimates of annual NDVI$_{max}$ using Monte Carlo simulations. These new techniques and the uncertainty analysis are briefly described below and in greater detail in the Supplementary Methods.

There are systematic differences in NDVI among Landsat 5, 7, and 8 (Supplementary Fig. 1) and failure to address these differences can introduce artificial positive trends into NDVI time series that are based on measurements from multiple sensors[20,76,77]. Cross-calibration models have been developed for other biomes[20,76,77], but not for Arctic tundra. We initially explored cross-calibrating sensors using linear regression, but found non-linearities that led us to develop a novel approach using machine learning algorithms that calibrated Landsat 5/8 to Landsat 7. Specifically, for each sampling site, we (1) identified the years when both sensors (i.e., Landsat 5/8 and Landsat 7) collected imagery, (2) computed 15-day moving median NDVI across the growing season for each sensor using measurements pooled across years, and (3) then randomly selected NDVI from one 15-day period with at least five observations from both sensors. We then used data from 2/3rd of sites to train Random Forest models[78] that predicted Landsat 7 NDVI based on Landsat 5/8 NDVI, while withholding data from the other 1/3rd of sites for cross-validation. The models also accounted for potential seasonal and regional differences between sensors by including as covariates the midpoint of each 15-day period (day of year) and the spatial coordinates of each site. We fit the Random Forest models using the ranger package[79] in R and evaluation of the models showed they had high predictive capacity ($r^2 \approx 0.97$) as well as low root mean-squared error and bias (Supplementary Fig. 1 and Supplementary Table 1). We therefore applied these models to cross-calibrate NDVI among sensors at the full set of sampling sites.

We inferred tundra greenness using estimates of annual NDVI$_{max}$ derived from the Landsat satellites; however, raw estimates of NDVI$_{max}$ are sensitive to the availability and seasonal timing of clear-sky measurements acquired each summer, particularly when few measurements are available. There are typically few clear-sky summer measurements at each Arctic sampling site during the 1980s and 1990s when only Landsat 5 was operating; however, observations became increasingly available during the 2000s following the launches of Landsat 7 and 8 (Supplementary Fig. 2). For instance, there was a median of 0, 2, 4, and 7 clear-sky summer scenes per sampling site in 1985, 1995, 2005, and 2015. As the number of clear-sky measurements increases, so does the likelihood of acquiring a measurement during the period of peak summer greenness. Consequently, we found that raw estimates of NDVI$_{max}$ increased asymptotically with the number of clear-sky measurements available each summer (Supplementary Fig. 3), which introduced a spurious positive trend into raw NDVI$_{max}$ time series given the increase in observations through time. We therefore developed a phenology-based approach to more reliably estimate NDVI$_{max}$ when few clear-sky summer measurements were available. Our approach involved modeling seasonal land surface phenology at each site for every 17-year period between 1985 to 2016 and then predicting annual NDVI$_{max}$ using individual summer measurements in tandem with information on phenology during the corresponding period

(Supplementary Fig. 4). The Landsat record is limited in much of the Arctic prior to the 2000s, thus using a 17-year window allowed us to pool measurements across this era of sparse observations when estimating annual NDVI$_{max}$. Specifically, for each site we quantified land surface phenology from spring through fall by predicting daily NDVI using flexible cubic splines fit to all clear-sky measurements. We then estimated annual NDVI$_{max}$ at each site by adjusting individual summer measurements based on the timing of acquisition relative to peak summer greenness (i.e., NDVI$_{max}$). A conceptually similar approach was previously used to examine inter-annual variability in the start and end of the growing season in deciduous forests of eastern North America[80]. Subsequent assessment showed that our modeled estimates of annual NDVI$_{max}$ were less biased than raw estimates of annual NDVI$_{max}$ when few summer measurements were available (Supplementary Fig. 3).

Several sources of uncertainty affect estimates of annual NDVI$_{max}$ and thus we propagated uncertainty into subsequent analyses using Monte Carlo simulations. This involved generating $10^3$ simulations of the annual NDVI$_{max}$ time series for every sampling site. For each simulation, we randomly varied measurements of red and near-infrared reflectance by up to ±7%, 5%, or 3% depending on whether measurements were from Landsat 5, 7, or 8, respectively[81,82]. We then estimated NDVI using each perturbed measurement of red and near-infrared reflectance. Afterwards, we cross-calibrated NDVI among sensors with a unique set of Random Forest models. Next, we estimated annual NDVI$_{max}$ at each site by fitting cubic splines of varying smoothness, implemented by randomly varying the smoothing parameter over a range of reasonable values (spar = 0.68–0.72). Overall, this process propagated several important sources of error and uncertainty into subsequent analysis, thus allowing us to more rigorously estimate greening and browning trends across the Arctic.

**Assessing Landsat NDVI$_{max}$ trends**. We assessed temporal trends in annual Landsat NDVI$_{max}$ during recent decades using measurements from sampling sites across the Arctic. We first excluded sampling sites that were barren (mean NDVI$_{max}$ < 0.10) or had short measurement records (<10 years) and then assessed temporal trends in NDVI$_{max}$ for each remaining sampling site ($n = 41,884$) as well as after averaging NDVI$_{max}$ time series across sampling sites in each bioclimatic zone and the Arctic. Justification for the sample size of 50,000 locations is provided in the Supplementary Methods (Supplementary Table 4 and Supplementary Fig. 5). We evaluated each time series for the presence of a monotonic trend using a rank-based Mann–Kendall trend test[83] and determined the slope of each time series using a non-parametric Theil–Sen slope estimator[84] as implemented using the zyp package[85] in R[86]. This approach for robust trend assessment accounts for potential temporal autocorrelation and has been used in prior studies that evaluated changes in NDVI at high latitudes[18,41]. We classified sites with a positive NDVI$_{max}$ trend ($\alpha = 0.10$) as greening or a negative NDVI$_{max}$ trend ($\alpha = 0.10$) as browning. Furthermore, we accounted for how trends were affected by uncertainty in estimates of annual NDVI$_{max}$ by computing every trend using each of the $10^3$ Monte Carlo simulations of annual NDVI$_{max}$ at every sampling site. We computed the median percentage of sites that greened or browned across all simulation and derived 95% confidence intervals using the 2.5th and 97.5th percentile of all simulations

**Summer air temperature data sets and analyses**. We assessed recent changes in summer air temperatures across the Arctic, as well as inter-annual covariation between summer temperatures and tundra greenness. Specifically, we characterized cumulative summer heat load using the SWI[28] derived from an ensemble of five global temperature data sets[87–91] re-gridded at 50 km resolution (Supplementary Table 5). The SWI is computed as the annual sum of mean monthly air temperatures exceeding 0 °C and is commonly used as an indicator of cumulative heat load in the Arctic[16,17,28]. Estimates of annual SWI differ among temperature data sets and thus to account for this uncertainty we performed a series of Monte Carlo simulations ($n = 10^3$). For each simulation, we generated a stack of annual synthetic SWI rasters built by randomly selecting grid cell values from the five temperature data sets. In other words, each grid cell of a synthetic raster was assigned a value for SWI that was randomly selected from the corresponding grid cell of one of the five temperature data sets. We then used this collection of synthetic SWI rasters to assess temporal trends in SWI as well as correlations between SWI and NDVI$_{max}$.

We assessed changes in summer temperatures using the synthetic SWI raster data sets and non-parametric trend tests in a Monte Carlo uncertainty framework. Specifically, for each of the $10^3$ Monte Carlo simulations, we evaluated SWI trends from 1985 to 2016 and 2000 to 2016 using non-parametric Mann–Kendall trend tests and Theil–Sen slope estimators as implemented by the zyp package[85] in R[86]. We assessed SWI trends for each $50 \times 50$ km$^2$ grid cell and Landsat sampling site, as well as after averaging SWI among grid cells in each bioclimatic zone and across the Arctic domain (Supplementary Table 6). We report the median change across all simulations as our best estimate of each trend and a 95% confidence interval computed from the 2.5th and 97.5th percentiles of these simulations.

We assessed the temporal correspondence between annual Landsat NDVI$_{max}$ and SWI from 1985 to 2016 and 2000 to 2016 at multiple spatial scales using rank-based $r_s$ in a Monte Carlo uncertainty framework. Specifically, we computed NDVI$_{max}$–SWI correlations for individual sampling sites and after averaging annual NDVI$_{max}$ and SWI time series among sites in each bioclimatic zone and

across the Arctic domain. Moreover, we assessed $NDVI_{max}$–SWI correlations using current and 2-year average SWI, as well as after linearly detrending the time series. Uncertainty in $NDVI_{max}$ and SWI can influence their association and thus we evaluated each correlation $10^3$ times by randomly pairing Monte Carlo simulations of each metric. We present the median $r_s$ of all simulations as our best estimate for each $NDVI_{max}$–SWI correlation and report a 95% confidence interval derived from the 2.5th and 97.5th percentile of all $r_s$ simulations. The $NDVI_{max}$–SWI correlations for each zone are summarized in Supplementary Table 7, while spatial patterns of these correlations are summarized in Supplementary Fig. 6.

**Evaluating potential drivers of changes in tundra greenness.** To explore potential drivers of changes in tundra greenness among sampling sites, we constructed Random Forest models[92] to predict the $NDVI_{max}$ trend class from 2000 to 2016 (i.e., browning, no trend, greening) based on environmental characteristics related to climate, permafrost, land cover, fire, and topography (Supplementary Table 8). We focused on 2000 to 2016 (rather than 1985 to 2016) given the more extensive spatial cover of Landsat and greater availability of predictor data sets during more recent years. Time series predictors included the ensemble SWI and minimum summer soil moisture[93] from 2000 to 2016, as well as permafrost extent[94], annual mean soil temperature (1 m depth)[95], and annual maximum active layer thickness[96] from 2003 to 2016. The permafrost data sets did not extend before 2003. We included both the linear change over time and model-fit starting value as predictors. Additional predictors included thermokarst vulnerability[97], ESA land cover[98], MODIS burned area (2001–2016)[54], and five topographic predictors (elevation, slope, aspect, topographic roughness, topographic position) derived from the TanDEM-X 90m Digital Elevation Model[99] (© DLR 2020). All together, we included 20 predictor variables in the Random Forests.

We constructed a separate Random Forest model for each of the $10^3$ Monte Carlo simulations. For every simulation, we classified the $NDVI_{max}$ trend at each sampling site as browning, no trend, greening based on the slope and significance ($\alpha = 0.10$) of $NDVI_{max}$ change from 2000 to 2016. The frequency of each trend class was highly skewed towards sites with no trend or greening and thus we balanced the sample size among trend class by determining the number of browning sampling sites and then randomly selecting the same number of no trend and greening sampling sites. We then screened highly correlated variables ($r > 0.75$) by computing pair-wise correlations and removing the variable with highest average absolute correlation. Next, we randomly partitioned the data set into sets for model training (2/3rd) and evaluation (1/3rd), and then repeatedly fit (i.e., tuned) Random Forest models to optimize out-of-bag classification accuracy by varying the number of variables assessed at each tree node. We selected the Random Forest model with the highest out-of-bag classification accuracy and then re-assessed the classification accuracy using the data withheld for model evaluation (Supplementary Tables 9 and 10). Last, we computed variable importance based on the mean decrease in accuracy metric and generated partial dependency plots to assess how class-specific classification probabilities varied across the range of each predictor while holding all other predictors at their average value. Model construction and evaluation were accomplished using functions from the randomForest[78], caret[100], and pdp[101] packages in R.

**Comparisons between Landsat $NDVI_{max}$ and plant productivity.** To aid in interpreting Landsat $NDVI_{max}$ trends, we compared $NDVI_{max}$ with three metrics of tundra plant productivity derived from field measurements at sites across the Arctic (Supplementary Figs. 1 and 7). The metrics of annual plant productivity included graminoid ANPP (g dry matter m$^{-2}$ year$^{-1}$), shrub RWIs (unitless), and ecosystem GPP (g C m$^{-2}$ year$^{-1}$). For each comparison, we incorporated uncertainty in both remote sensing and field data sets using Monte Carlo simulations. We briefly describe each comparison below and include additional details in the Supplementary Methods.

We assessed the temporal correspondence between annual Landsat $NDVI_{max}$ and graminoid ANPP from 1990 to 2017 on Bylot Island in northern Canada (Supplementary Fig. 7a)[8]. Graminoid ANPP has been monitored each year as part of a long-term study focused on Arctic food chains and was quantified by annually clip harvesting live aboveground biomass in 11 to 12 quadrats (20 × 20 cm$^2$) in the study site. We developed annual $NDVI_{max}$ time series for four subsites and then assessed the relationship between annual median $NDVI_{max}$ and ANPP from 1990 to 2017 using $r_s$ in a Monte Carlo uncertainty framework ($n = 10^3$ simulations). Each simulation randomly perturbed both $NDVI_{max}$ and ANPP data sets and utilized data from a random subset (90%) of years. We also explored multi-year and lagged relationships between $NDVI_{max}$ and ANPP.

We assessed the temporal correspondence between annual Landsat $NDVI_{max}$ and shrub growth using 22 shrub RWI chronologies from sites in six Arctic countries (Supplementary Fig. 7b and Supplementary Table 11). The shrub RWI chronologies are a proxy for inter-annual variability in shrub productivity and in some cases may co-vary with broader plant community productivity[102]. We used new and archived measurements of alder (*Alnus* spp.), willow (*Salix* spp.), and birch (*Betula* spp.) annual ring width from independent projects[10,12,34,67], including measurements previously collated as part of the ShrubHub shrub ring database[42]. We generated a detrended and standardized median shrub RWI chronology for each shrub genera at a site using the dplR[103] package in R. We also developed annual detrended Landsat $NDVI_{max}$ ($NDVI_{max-dt}$) time series using

observations from a 100 m radius area around each sampling location. We then assessed the temporal correspondence between $NDVI_{max-dt}$ and each shrub RWI chronology using Spearman's correlations in a Monte Carlo uncertainty framework ($n = 10^3$ simulations). Each simulation randomly perturbed both $NDVI_{max}$ and shrub RWI data sets and utilized data from a random subset (90%) of years.

We assessed the spatial correspondence between median annual Landsat $NDVI_{max}$ and ecosystem GPP across 11 eddy covariance flux towers located in Arctic tundra of Greenland, Russia, and the USA (Supplementary Fig. 7c and Supplementary Table 12). Four of the flux towers were part of the Arctic Observing Network[104,105] and seven of the flux towers were part of the FLUXNET Network (FLUXNET2015 Tier 1)[106]. Annual ecosystem GPP was estimated at each flux tower by first measuring net ecosystem exchange (NEE) and then partitioning NEE into GPP and ecosystem respiration ($R_{eco}$) using modeled relationships between $R_{eco}$ and night-time temperatures (NEE = GPP − $R_{eco}$)[107]. We acquired annual gap-filled estimates of GPP from FLUXNET and half-hourly gap-filled estimates of GPP from AON that we aggregated to an annual time step (g C m$^{-2}$ year$^{-1}$). We generated annual Landsat $NDVI_{max}$ time series for each flux tower using summer observations from a 100 m radius area around each flux tower. We then computed median annual $NDVI_{max}$ and GPP by site and assessed their covariation using $r_s$ in a Monte Carlo uncertainty framework ($n = 10^3$ simulations). Each simulation randomly perturbed both $NDVI_{max}$ and GPP data sets and utilized data from a random subset (90%) of years.

**Data handling and visualization.** We acquired Landsat data using Python[74] and generated maps using ArcGIS (Redlands, CA), but otherwise handled and visualized data using R[86] with a suite of add-on packages. Specifically, we processed geospatial data using raster[108], rgdal[109], and maptools[110]. Furthermore, we handled data using data.table[111], dplyr[112], and tidyr[113], and visualized data using lattice[114], ggplot2[115], and ggpubr[116]. All package versions are provided in the Reporting summary.

**Reporting summary.** Further information on experimental design is available in the Nature Research Reporting Summary linked to this paper.

## Data availability

The data that support the findings of this study are available from the following sources: The United States Geologic Survey Landsat 5, 7, and 8 Surface Reflectance data are available from Google Earth Engine. The CRU TS4.01: Climatic Research Unit (CRU) Time Series (TS) version 4.01 data are available from the Center for Environmental Data Analysis with identifier https://doi.org/10.5285/58a8802721c94c66ae45c3baa4d814d0. The Terrestrial Air Temperature: 1900–2017 Gridded Monthly Time Series (V 5.01) data are available from the University of Delaware, http://climate.geog.udel.edu/~climate/html_pages/download.html#T2017. The Land-Ocean Temperature Index ERSSTv5 data are available from the NASA Goddard Institute for Space Studies, https://data.giss.nasa.gov/pub/gistemp/GHCNv3/gistemp1200_ERSSTv5.nc.gz. The Monthly Land + Ocean Average Temperature with Air Temperatures at Sea Ice data are available from Berkeley Earth. The HadCRUT4 hybrid with UAH data are available from the University of York, https://www-users.york.ac.uk/~kdc3/papers/coverage2013/had4_short_uah_v2_0_0.nc.gz. The TerraClimate data are available from the University Corporation for Atmospheric Research, http://thredds.northwestknowledge.net:8080/thredds/catalog/TERRACLIMATE_ALL/data/catalog.html. The Arctic Circumpolar Distribution and Soil Carbon of Thermokarst Landscapes (2015) data are available from the Oak Ridge National Laboratory with identifier https://doi.org/10.3334/ORNLDAAC/1332. The ESA Climate Change Initiative Permafrost extent, active layer thickness, and ground temperature data are available from the Center for Environmental Data Analysis with identifiers https://doi.org/10.5285/c7590fe40d8e44169d511c70a60ccbcc, https://doi.org/10.5285/1ee56c42cf6c4ef698693e00a63795f4, and https://doi.org/10.5285/c7590fe40d8e44169d511c70a60ccbcc, respectively. The ESA Climate Change Initiative Land cover data are available from the Catholic University of Louvain, http://maps.elie.ucl.ac.be/CCI/viewer/download.php. The MODIS/Terra + Aqua Burned Area Monthly L3 Global 500m data are available from the Land Processes Distributed Active Archive Center, https://lpdaac.usgs.gov/products/mcd64a1v006/. The TanDEM-X 90m Digital Elevation Model data are available from the German Aerospace Center, https://geoservice.dlr.de/web/dataguide/tdm90/#access. The graminoid productivity data are available upon reasonable request from G.G. The shrub ring-width data are available from the (1) Polar Data Catalog with identifier, https://www.polardata.ca/pdcsearch/PDCSearchDOI.jsp?doi_id=12131, (2) the Arctic Data Center with identifiers https://doi.org/10.18739/A28Q18 and https://doi.org/10.18739/A24X0Q, and (3) the National Center for Environmental Information with identifiers https://www.ncdc.noaa.gov/paleo/study/29754, https://www.ncdc.noaa.gov/paleo/study/29752, and https://www.ncdc.noaa.gov/paleo/study/29753. Additional shrub ring-width data are available upon reasonable request from B.C.F. and B.V.G. The gross primary productivity data are available from the Arctic Observing Network, http://aon.iab.uaf.edu/data_access. Additional primary productivity data are available from Fluxnet with identifiers https://doi.org/10.18140/FLX/1440182, https://doi.org/10.18140/FLX/1440067, https://doi.org/10.18140/FLX/1440073, https://doi.org/10.18140/FLX/1440181, https://doi.org/10.18140/FLX/1440222, https://doi.org/

10.18140/FLX/1440224, and https://doi.org/10.18140/FLX/1440223. The Landsat data sets generated as part of this project will be publicly archived with the Oak Ridge National Laboratory Distributed Active Archive Center for Biogeochemical Dynamics.

## Code availability

All code from this analysis is publicly archived on the lead authors GitHub.

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

## Acknowledgements

This work was supported by the National Aeronautics and Space Administration (NASA) Arctic Boreal Vulnerability Experiment (ABoVE) grants NNX17AE44G and 80NSSC19M0112 to S.J.G., NASA Carbon Cycle Science grant NNX17AE13G to S.J.G, and National Science Foundation (NSF) Arctic Natural Sciences grant 1661723 to R.D'A., L.A.-H., and S.J.G. Additional support provided by NSF Partnerships for International Research and Education grant 1743738 and NSF Division of Atmospheric and Geospace Sciences grant 1502150 to R.D'A. B.C.F. was supported by the Academy of Finland (grant 256991), the Joint Program Initiative Climate (grant 291581), and the European Commission Research and Innovation Action (grant 869471). T.K. was supported by the Academy of Finland (grant 330319). M.M.-F. and I.M.-S. acknowledge support from the United Kingdom National Environmental Research Council (grants NE/L011859/1 and NE/M016323/1, respectively). B.V.G. was supported by the Joint Fire Science Program (grant 16-1-01-8). L.A.-H. acknowledges support from NSF Polar Programs (grant 15-04134) and the Lamont-Doherty Earth Observatory Climate Center. G.G. acknowledges support from the Natural Science and Engineering Research Council of Canada, Environment and Climate Change Canada, the network of center of excellence ArcticNet, and the Polar Continental Shelf Program. Landsat Surface Reflectance products were provided courtesy of the U.S. Geological Survey. This work used eddy covariance data acquired and shared by the FLUXNET community, with additional eddy covariance data provided by the Institute of Arctic Biology, University of Alaska Fairbanks, based on the work supported by the National Science Foundation (grant 1107892). Computational analyses were run on Northern Arizona University's Monsoon computing cluster, funded by Arizona's Technology and Research Initiative Fund.

## Author contributions

L.T.B. and S.J.G. designed the study with input from P.J., B.C.F., M.M.-F., and I.M.-S. Acquisition and processing of Landsat data were aided by R.M. and P.B. Acquisition and processing of climate data were aided by P.J. Field data and ideas were contributed by G.G., B.C.F., M.M.-F., T.K., L.A.-H., B.V.G., P.Z., and R.D'A. The development of Landsat processing procedures, analyses, and writing were led by L.T.B., with input and edits from all authors.

## Competing interests

The authors declare no competing interests.
