## [Peer Review File · Nature Communications]

Reviewers' comments:

Reviewer #1 (Remarks to the Author):

Review of the paper entitled " Landsat satellites reveal widespread greening in the Arctic tundra biome during the last three decades" by Berner et al.

1) The paper lacks originality, as pointed by the authors themselves the greening has been shown by other satellites at coarser spatial resolution.

2) The authors brought up the fact that high spatial resolution is more adapted to study those questions but they fail to recognize/address that temporal resolution (daily or twice daily observations) is as much or even more important to capture the phenology of vegetation, with Landsat data at 16 days interval (at the best) factoring in clouds and other issues it is really challenging or impossible to capture parameters such as the NDVImax during the growing season.

3) They also pointed out that three satellites are easier to cross-calibrate (removing spurious trend due to satellite change) than "16 separate satellites", however they encountered some issues that they fix using techniques that are not traceable (machine learning), in my opinion those fixes are not only addressing potential cross-calibration issues but other issues like temporal sampling, quality of the data (for example in the surface reflectance products). If there is issue with the cross-calibration, they should be either reported to the landsat team and/or accounted for in the error analysis/error budget. By the way, this paper does not provide any error analysis on the NDVImax derived.

4) Finally, it is quite surprising that only (50,000 samples) were analyzed, roughly a 200x200 image (that is not even a full Landsat scene). Given the relatively simple analysis conducted and nowadays computer capability, it should be possible to analyze the full dataset.

So the lack of originality, basic problems with the approaches used in the analysis (2,3) un-adequate sample size (4), I have to reject that paper.

Reviewer #2 (Remarks to the Author):

Review of „Landsat satellites reveal widespread greening in the Arctic tundra biome during the last three decades" by Logan T. Berner et al.

The study uses Landsat imagery to analyse greening and browning trends in Arctic vegetation. This is an important publication because it provides for the first time a pan-arctic view on greening trends based on high-resolution satellite data. Previously, lower resolution data from MODIS or AVHRR was used for such analyses, which partly provided conversely results. Remote sensing of vegetation in arctic regions is complicated by the short growing season, the high sun zenith angles, frequent cloud cover and the heterogeneity of the surface. Although effects of surface heterogeneity can be accounted by using high-resolution satellite data like from Landsat, analyses of this data are more affected by the shorter revisit time (and hence less frequent observations) than from medium-resolution satellites. Here the authors combined a couple of statistical and machine learning methods to enable the analysis of greening trends. For example, they used random forest to calibrate Landsat 5 and 8 data to Landsat 7 and used a flexible time series smoothing technique to estimate the phenological cycle of NDVI and derive NDVImax which avoids potential errors in estimating NDVImax from the infrequent sampling of Landsat.

The paper is well written and clear. The methods and supplement provide enough detail to understand the methods and the provided code potentially allows to reproduce the results. The authors should

consider to later compile all the functions for the processing of Landsat data in an R package as this might provide a basis for further analyses of Landsat data in northern ecosystems. This is a great analysis and paper.

Minor comments

Page 9: Please provide an additional statement (and supplementary figure) about the covariation of NDVI_max trends and NDVI_max-SWI correlations. This is important to understand the following questions:

- Do all sites with greening also have positive NDVI_max-SWI correlations?
- Do the sites with browning have negative NDVI_max-SWI correlations?
- Which regions without greening have still positive NDVI_max-SWI correlations?
- This could help to understand the potential importance of other environmental controls, which are already discussed at page 13.

Figure 1 (b and c): It is not easy to identify the number of sample sizes with the continuous blue colour palette. I suggest to use class breaks (e.g. [1-5) [5-10) [10-5) ... [25-30)) instead to improve the visualization (and potentially a different colour).

Figure 3: I suggest to add a colour legend to the plot to make it is easier to read the figure. Please clarify in the caption if the thresholds of $\alpha = 0.05$ and 0.10 for the dark and light shades correspond to the significance level of the trends and correlations or to absolute trend changes and correlations.

Matthias Forkel, Dresden, Germany, 6. January 2020

Reviewer #3 (Remarks to the Author):

Berner and colleagues present the results of a study aimed at using long-term and high resolution satellite data to track changes in maximum 'greenness' (quantified in terms of the normalized difference vegetation index) over the arctic. They find increased greening at 45% of the locations studied, and correlate the greening with historic increases in air temperature. The results suggest that the land surface of high-latitudes is greening, or at least ~50% of it is.

The manuscript is very well written, and the analysis clearly described. The strength of the analysis is that the authors are working with Landsat data, which provides retrievals at very high spatial resolution (30m). Previous studies focused on high latitude greening estimated from remote sensing, of which there are many, used coarser resolution (~8km) data. Working with such high resolution data is computationally challenging, as evidenced by the large amount of code submitted with the manuscript. This is a very commendable effort and adds significant value to the manuscript.

My main concerns regard some of the methods employed, and the novelty.

In particular, the spatial resolution of the Landsat data comes at the expense of temporal coverage. As the authors note and develop methods to correct for, the sparse temporal coverage also has a temporal dynamic. I.e. The further back you go in time the fewer retrievals there are. With few retrievals it becomes difficult to estimate the maximum NDVI in each year. In order to estimate maximum NDVI each year the authors therefore use a model informed by the mean seasonal NDVI over all years. This could potentially introduce some bias, as the length of the seasonal cycle has been extending over the decades due to warming. By imposing a mean seasonal cycle (see Fig. S4), the authors are potentially overestimating the maximum NDVI in the early parts of the timeseries, as the actual growing season was likely shorter than the mean growing season that they impose. This could potentially lead to an underestimation of the greening trend in the authors analysis. The authors

could test this by imposing a temporal trend in the mean seasonal cycle, through as far as I know the true change is unknown due to the sparsity of retrievals. Note that the text frequently talks about pan-arctic but really much of eastern Eurasia has no retrievals pre-2000, and western Eurasia has very spotty retrievals for much of the prior decades also.

The correlation with the SWI anomaly is interesting but potentially misleading. What the authors present as an anomaly is in fact a normalized change, in that the time series is not detrended. As can be seen from Fig. 2c there is no relationship between NDVI and SWI anomalies within decades (with the exception perhaps of the 2010s), it is only across decades that the relationship emerges due to the lack of detrending. What this analysis shows therefore is that both NDVI and SWI are trending, but it does not show that one is sensitive to the other (as claimed by the authors). In order to show that NDVI is sensitive to changes in SWI then you should detrend both NDVI and SWI. Otherwise, you should revise the text to say that both are trending up, and although there is no correlation between one and the other on an interannual (or multi-annual) scale there is reason to believe they are related over longer timescales.

My final concern is one of novelty. There have been many many studies calculating trends in NDVI from remote sensing data. The authors motivate the study by the fact that previous studies of greening focus on coarser resolution retrievals, and the 30m resolution used here is needed to accurately quantify trends. But I did not really see where the strength of this high resolution led to conclusions that were not obtainable with the coarser resolution products. In the end the authors aggregate their results to three very coarse zones. The fine resolution leads to spatial and temporal data sparsity, so it could equally be argued that the coarser resolution satellites are better as they cover larger areas and have more frequent data availability. Although the authors did a great job of using the fine resolution to compare to field data, I was disappointed that the emphasis on fine resolution did not lead to any new conclusions, and was not used to help understand the large differences between previously published results using coarser resolution data. Those differences are discussed in detail in a recent paper by one of the authors, Myers-Smith, published in Nature Climate Change (DOI: 10.1038/s41558-019-0688-1, not cited). Showing how the fine resolution leads to more defensible conclusions would be a great addition to this manuscript.

Minor comments:

For future submissions note that line numbers are always appreciated.

Page 2: "resolve disparate finding(s)"

Page 2: The references to the field studies are somewhat out of date. See for example: DOI: 10.1002/ecm.1351

Page 7: The correlation between NDVImax and SWI anomalies mostly seems to be driven by the increasingly positive anomalies in SWI over time. Ie. As NDVI trends then it's not surprising that there is a correlation with another trending variable. But should the SWI anomalies be trending?

Responses to Reviewers

Reviewer #1 (Remarks to the Author):

Review of the paper entitled “ Landsat satellites reveal widespread greening in the Arctic tundra
biome during the last three decades” by Berner et al.

1) The paper lacks originality, as pointed by the authors themselves the greening has been shown
by other satellites at coarser spatial resolution.

Author response: Current understanding of pan-Arctic trends in tundra greenness since the 1980s
comes exclusively from the AVHRR satellites. These meteorological satellites provide crucial
long-term perspective on global land surface change; however, these satellites were not designed
for monitoring vegetation, suffer serious issues with cross-sensor calibration, and provide
observations at a spatial scale (~ 8 km x 8 km) that limits not only validation against field
measurements and but also inference about drivers of landscape change (Guay et al. 2014,
Pattison et al. 2015, Myers-Smith et al. 2020). Our team has extensively assessed changes in
high-latitude vegetation using the AVHRR satellites (e.g., Bunn et al. 2005, Goetz et al. 2005,
Bunn and Goetz 2006, Beck and Goetz 2011, Berner et al. 2011, Berner et al. 2013, Guay et al.
2014); experience that gives us appreciation for both the capabilities and limitations of AVHRR.
These limitation have led to recent calls for studies to utilize higher resolution Landsat satellites
to assess tundra greening and browning, including by the National Academies of Sciences (2019)
and in a new review paper published by Nature Climate Change (Myers-Smith et al. 2020).

Our current study directly responses to these calls by providing the first ever assessment
of (1) changes in tundra greenness, (2) potential drivers of change, and (3) links between tundra
greenness and field measurements across the Arctic using Landsat observations. While our
analysis provides critical independent confirmation of biome-scale *greening* observed by
AVHRR, it also illustrates notable difference in the spatial extent and regional patterns of
*greening* and *browning* among satellites. Furthermore, the changes in tundra greenness that we
observed using Landsat are supported by comparisons with three unique metrics of tundra plant
productivity measured at over 30 field sites in the Arctic; a scope of satellite – field data
comparisons that has never previously been seen in this biome. Additionally, our revised analysis
now includes a robust uncertainty analysis and demonstrates that spatial patterns of greening and
browning were affected by changes in summer temperatures, ground temperatures, and soil
moisture. Altogether, our analysis provides critical independent support for biome-scale *greening*
previously observed by the AVHRR satellites, while also highlighting not only regions with
disagreement among satellites but also important factors contributing to tundra greenness
dynamics across the rapidly warming Arctic. Our methodological advances also help lay a
foundation for future Landsat-based assessments of environmental change in the Arctic and
beyond.

2) The authors brought up the fact that high spatial resolution is more adapted to study those
questions but they fail to recognize/address that temporal resolution (daily or twice daily

observations) is as much or even more important to capture the phenology of vegetation, with
Landsat data at 16 days interval (at the best) factoring in clouds and other issues it is really
challenging or impossible to capture parameters such as the NDVImax during the growing
season.

Author response: The AVHRR satellites provide an invaluable record of land surface change
since the 1980s; however, we would be remiss to rely *exclusively* on these satellites for assessing
critical environmental change across Earth's surface. The AVHRR satellites provide daily or
twice daily observations that are generally used to develop bi-monthly NDVI data at 1/12 °
spatial resolution (~8 km) (Pinzon and Tucker 2014), whereas the Landsat satellites provide
observations about every 8 to 16 days (depending on time period) at 30 m spatial resolution.
There are perhaps 10 to 30 times fewer Landsat observations during a given period; however,
these observations are about 71,000 times higher spatial resolution and measure wavelengths that
are more suitable for monitoring vegetation. The vastly superior spatial resolution and more
finely tuned spectral bands makes it possible to focus on tundra vegetation while carefully
filtering out standing water, snow patches, and clouds, all of which can impact NDVI trends
(e.g., Myers-Smith et al. 2020). Nevertheless, the low frequency of Landsat observations
presents a notable challenge when trying to characterize tundra vegetation phenology in the
Arctic where the growing season is short and often cloudy.

Several studies have simply used Landsat data from a specific seasonal window (e.g.,
July – August) when assessing trends in summer NDVI in portions of the Arctic (e.g., Ju and
Masek 2016, Nitze and Grosse 2016); however, this approach is inadequate for a pan-Arctic
assessment where the duration of the growing season is highly variable across the domain. In
responding to this challenge, we developed and evaluated a new phenology-based approach to
estimate annual NDVI_{max} using Landsat data. We devote almost six pages of supplemental
material to describing not only the challenges associated with sparse Landsat observations but
also our approach to overcoming these challenges (*Section 1.5. Modeling maximum summer*
*NDVI using Landsat*). To help address the reviewer's concern, we also added the following text
to the introduction and discussion to better highlight the challenge of using Landsat to assess
changes in vegetation phenology in the Arctic:

[Introduction] The Landsat satellites provide 30 m resolution observations that more
closely match the scale of field measurements and ecological change than AVHRR
observations. However, higher spatial resolution means that each location is observed
few times each growing season, particularly since the growing season is short and often
cloudy in the Arctic. It is therefore challenging to characterize changes in vegetation
phenology in this region (Karlsen et al. 2018).

[Discussion] Compared to AVHRR or MODIS, high resolution Landsat observations
may be especially advantageous when assessing vegetation dynamics in highly
heterogeneous tundra landscapes (Myers-Smith et al. 2020). Nevertheless, the paucity of
Landsat observations in parts of the Arctic (e.g. eastern Eurasia) prior to 2000 constrain
the utility of these sensors, as does the lower frequency of observations acquired each
summer. We partially addressed the effects of low observation frequency using a novel

approach to modeling annual $NDVI_{max}$ with site-specific information on land surface
phenology.

3) They also pointed out that three satellites are easier to cross-calibrate (removing spurious
trend due to satellite change) than “16 separate satellites”, however they encountered some issues
that they fix using techniques that are not traceable (machine learning), in my opinion those fixes
are not only addressing potential cross-calibration issues but other issues like temporal sampling,
quality of the data (for example in the surface reflectance products). If there is issue with the
cross-calibration, they should be either reported to the landsat team and/or accounted for in the
error analysis/error budget. By the way, this paper does not provide any error analysis on the
$NDVI_{max}$ derived.

Author response: The fact that the three Landsat satellites suffer issues with cross-sensor
calibration further highlights the challenges associated with cross-calibrating so many AVHRR
sensors. Members of the Landsat Science Team are aware of existing issues with cross-sensor
calibration and have published several papers documenting these discrepancies in other regions
(e.g., Ju and Masek 2016, Roy et al. 2016). Nevertheless, issues with cross-sensor calibration
persist. Rather than ignoring this critical issue or relying on cross-calibration regression models
from other biomes, we instead developed a novel machine learning technique to address this
issue specifically for the tundra biome. While perhaps less “traceable” than a simple regression
model, the machine learning technique was nevertheless highly effective (e.g. $r^2 \approx 0.97$, bias <
0.0001 NDVI) and able to account for the non-linear relationship observed between NDVI from
Landsat 7 and 8. To better illustrate the need for additional cross-sensor calibration, we added a
figure to the supplemental material showing the relationships between NDVI among sensors
before and after cross-sensor calibration (Fig. S1). We believe this is a reasonable approach to
address the issue of cross-sensor calibration and provide detailed description of the approach in
the supplemental material.

**Fig. S1** | There are systematic differences in NDVI among Landsat sensor. Landsat 7 NDVI
 compared with raw and cross-calibrated (a) Landsat 5 NDVI and (b) Landsat 8 NDVI. Note that
 raw Landsat 5 NDVI was consistently lower than Landsat 7 NDVI, which was consistently lower
 than Landsat 8 NDVI (left columns). This can introduce an artificial positive trend in composite
 NDVI time series. This issue was obviated by further cross-sensor calibration using machine
 learning (right columns). Each data point is an estimate of 15-day median NDVI computed from
 observations acquired during the years of overlap between pairs of sensors at each site. Each site
 contributes a single data point with the 15-day period select at random from available periods
 during summers with at least five observations. The diagonal orange lines show 1:1
 relationships.

We also developed a novel and effective technique to help address effects of “temporal
 sampling” on estimates of $NDVI_{max}$ and, as part of this process, evaluated error associated with
 our estimates of $NDVI_{max}$. The error assessment is shown in Figure S3 (referenced in the
 methods section) and described on page eight of the supplemental material. Briefly, we

computed observed $NDVI_{max}$ at sites with at least 11 Landsat scenes acquired in a summer and
then examined how both raw (i.e., simple max) and phenology-based estimates of $NDVI_{max}$
differed from observed $NDVI_{max}$ as a function of scene availability. As reported in the
supplemental, “This assessment showed that raw estimates of $NDVI_{max}$ increase asymptotically
until there are at least seven Landsat scenes acquired during a summer, after which estimates of
$NDVI_{max}$ change little with increasing scene availability. On the other hand, our phenologically-
corrected estimates of $NDVI_{max}$ change little with increasing scene availability, though the
uncertainty of the estimates decreases with increasing scene availability... These comparisons
highlight that (1) estimates of annual $NDVI_{max}$ are sensitive to the number of available scenes
and that (2) our phenological correction can provide less biased estimates of annual $NDVI_{max}$
when few Landsat scenes are available from a growing season.”

We do acknowledge the potential to better address uncertainty in our analysis and
therefore we substantially revised the entire analysis to now include a Monte Carlo uncertainty
framework. As part of this process, we now generate 10^3 simulations of the annual $NDVI_{max}$ time
series for each sample site. Each simulation involved (1) randomly permuting red and near-
infrared reflectance by sensor-specific uncertainty before computing NDVI, (2) calibrating
NDVI among sensors using unique random forest models, and (3) randomly permuting
parameters associated with fitting the phenological curves when estimating $NDVI_{max}$. For each
simulation we evaluated $NDVI_{max}$ trends, inter-annual relationships between $NDVI_{max}$ and
summer temperatures, and environmental drivers of trends in $NDVI_{max}$. Furthermore, we
extended the Monte Carlo analysis to include our assessment of temperature trends based on five
data sets, as well as our assessments of relationships between $NDVI_{max}$ and field measurements
of plant productivity. This major revision allows us to now robustly characterize and report
uncertainty (e.g., 95% Monte Carlo confidence intervals) in all aspects of our analysis.

4) Finally, it is quite surprising that only (50,000 samples) were analyzed, roughly a 200x200
image (that is not even a full Landsat scene). Given the relatively simple analysis conducted and
nowadays computer capability, it should be possible to analyze the full dataset.

Author response: Robust analysis of Landsat $NDVI_{max}$ trends wall-to-wall for every 30 m pixel
in the Arctic since 1985 is a great aspiration; however, this has yet to be accomplished by any
research team and is not necessary to draw robust inference about recent ecological change
across the pan-Arctic domain. The apparent simplicity of our analysis masks considerable
underlying complexity and computational intensity in the analytic approach, including
processing over 500 million multi-band measurements of surface reflectance sampled from
~75,000 Landsat scenes collected over 32 years. To assuage the reviewer’s concern about the
adequacy of the sample size, we performed an additional analysis and added the following
content to the supplemental material (*Section 1.6 Landsat $NDVI_{max}$ trend analysis*):

In this study, we assessed $NDVI_{max}$ trends using Landsat observations from
50,000 random sample sites in the Arctic. To evaluate the adequacy of this sample size,
we examined how estimates of two trend metrics varied as a function of sample size.
Focusing on 2000 to 2016, we computed the change in mean Arctic $NDVI_{max}$ and the

percentage of sites with positive, negative, or no trend ($\alpha = 0.10$) using sample sizes
 ranging from 10^2 to 4×10^4 sites. Specifically, we used samples sizes from 10^2 to 10^3 sites
 at intervals of 10^2 sites and then from 10^3 to 4×10^4 sites at intervals of 10^3 sites ($n = 49$
 bins total). For each Monte Carlo simulation ($n = 10^3$), we computed these trend metrics
 using random subsets of sites for each of the 49 sample size bins. We then computed the
 median and 95% confidence interval (CI) of each trend metric for every sample size bin.

This analysis revealed minimal differences in trend estimates across a 400-fold
 range in sample size (Table S1, Fig. S2). For instance, we estimated that the median
 increases in mean Arctic NDVI_{max} was 3.51%, 3.38%, or 3.35% whether based on 10^2 ,
 10^4 , or 4×10^4 sample sites. Moreover, we estimated that the median percentage of sites
 with a positive NDVI_{max} trend ($\alpha = 0.10$; “greening”) was 21.00%, 21.30%, and 21.30%
 based on 10^2 , 10^4 , or 4×10^4 sampling sites, while the median percentage of sites with a
 negative NDVI_{max} trend (“browning”) was 6.00% across all sizes. The width of the 95%
 CIs associated with these trend metrics asymptotically shrank, with about a 0.5% change
 between 10^4 and 4×10^4 sampling sites (Fig. S2c,d). This analysis illustrates the sample
 size is adequate for drawing robust inference about recent changes in tundra greenness
 across the Arctic.

**Table S1 | Effects of sample size on estimates of Landsat NDVI_{max} trends in the Arctic from**
 **2000 to 2016.** Trend characteristics include the total relative change in mean Arctic NDVI_{max} and
 the percentage of sites with a positive (“greening”), negative (“browning”), or no trend in
 NDVI_{max} ($\alpha = 0.10$). Each trend metric represents the median estimate from 10^3 Monte Carlo
 simulations and is accompanied by a 95% confidence interval (CI). Note that each median trend
 metric is quite stable across a 400-fold range in sample size and that the width of each 95% CI
 asymptotically decreases.

Number of sample sites	Change in mean Arctic NDVI (%)	Percent of sampling sites		
		Browning	Greening	No trend
10^2	3.51 [0.14, 6.43]	6.00 [2.00, 11.00]	21.00 [14.00, 30.00]	73.00 [63.00, 81.00]
10^3	3.34 [2.38, 4.35]	6.00 [4.60, 7.50]	21.30 [18.80, 23.80]	72.70 [70.00, 75.40]
10^4	3.38 [2.98, 3.77]	6.00 [5.50, 6.50]	21.30 [20.40, 22.20]	72.70 [71.90, 73.60]
4×10^4	3.35 [3.18, 3.51]	6.00 [5.70, 6.30]	21.30 [20.80, 21.80]	72.70 [72.30, 73.10]

**Fig. S2 | Effects of sample size on estimates of Landsat NDVI_{max} trends in the Arctic from**
 **2000 to 2016.** Trend metrics include (a) the relative change in mean Arctic NDVI_{max} (%) and (b)
 the percentage of sites with a positive (“greening”) or negative (“browning”) trend in NDVI_{max}
 ($\alpha = 0.10$). Solid lines depict median estimates from 10^3 Monte Carlo simulations while error
 bands depict 95% confidence intervals (CI). Changes in the width of the 95% CIs are shown in
 panels (c) and (d). Each simulation not only used random subsets of sites, but also NDVI_{max} time
 series generated with randomly permuted surface reflectance, cross-sensor calibration models,
 phenological-correction parameters.

Mapping Landsat NDVI_{max} trends wall to wall across the Arctic could provide additional insight
 into Arctic environmental changes, but is beyond the scope of our current study and should be
 explored as part of future research. In addition to its scientific merit, our study helps build the
 foundation for such future work by identifying and developing solutions to resolve critical
 methodological issues (e.g., cross-sensor calibration, sparsity of observations) required to
 perform a robust NDVI trend analysis for the pan-Arctic domain.

So the lack of originality, basic problems with the approaches used in the analysis (2,3) un-
adequate sample size (4), I have to reject that paper.

*Author response:* Our current understanding of pan-Arctic trends in tundra greenness since the
1980s comes exclusively from the AVHRR satellites, which were not designed for monitoring
vegetation and have a suite of other issues (e.g., Myneni et al. 1997, Goetz et al. 2005, Guay et
al. 2014). The Landsat satellites were designed for monitoring vegetation at high spatial
resolution and offer the only alternative to AVHRR satellites for such long-term assessments.
Never before now have the Landsat satellites been used to (1) evaluate pan-Arctic trends in
tundra greenness, (2) establish links between tundra greening and climate change, (3) or been
validated against three unique types of field measurements at over 30 sites spread across the
Arctic. We developed novel and effective techniques to address challenges with sparse
observations and cross-sensor calibration when utilizing the Landsat record and illustrate that our
sample size is adequate for assessing changes across the Arctic. Furthermore, our analysis now
includes robust uncertainty analyses based on Monte Carlo simulations, as well as closer
examination of factors contributing to spatial variability in NDVI trends across the Arctic. We
thus respectfully disagree with the reviewer's interpretation of our study. Instead, we believe that
our study makes a unique contribution to our understanding of recent Arctic environmental
change by following recent recommendations from the National Academies of Sciences (2019)
and over 40 members of the Arctic tundra research community (Myers-Smith et al. 2020).

**Reviewer #2 (Remarks to the Author):**

Review of "Landsat satellites reveal widespread greening in the Arctic tundra biome during the
last three decades" by Logan T. Berner et al.

The study uses Landsat imagery to analyse greening and browning trends in Arctic vegetation.
This is an important publication because it provides for the first time a pan-arctic view on
greening trends based on high-resolution satellite data. Previously, lower resolution data from
MODIS or AVHRR was used for such analyses, which partly provided conversely results.
Remote sensing of vegetation in arctic regions is complicated by the short growing season, the
high sun zenith angles, frequent cloud cover and the heterogeneity of the surface. Although
effects of surface heterogeneity can be accounted by using high-resolution satellite data like from
Landsat, analyses of this data are more affected by the shorter revisit time (and hence less
frequent observations) than from medium-resolution satellites. Here the authors combined a
couple of statistical and machine learning methods to enable the analysis of greening trends. For
example, they used random forest to calibrate Landsat 5 and 8 data to Landsat 7 and used a
flexible time series smoothing technique to estimate the phenological cycle of NDVI and derive
NDVImax which avoids potential errors in estimating NDVImax from the infrequent sampling
of Landsat. The paper is well written and clear. The methods and supplement provide enough
detail to understand the methods and the provided code potentially allows to reproduce the
results. The authors should consider to later compile all the functions for the processing of
Landsat data in an R package as this might provide a basis for further analyses of Landsat data in

northern ecosystems. This is a great analysis and paper.

Author response: We thank Dr. Forkel for reviewing our manuscript, recognizing the merits of
our study, and providing helpful feedback. We implemented the changes suggested by Dr. Forkel
and will consider compiling the functions into an R package in addition to making the code
publicly available on GitHub.

Minor comments

Page 9: Please provide an additional statement (and supplementary figure) about the covariation
of NDVI_{max} trends and NDVI_{max}-SWI correlations. This is important to understand the
following questions:

- Do all sites with greening also have positive NDVI_{max}-SWI correlations?

- Do the sites with browning have negative NDVI_{max}-SWI correlations?

- Which regions without greening have still positive NDVI_{max}-SWI correlations?

- This could help to understand the potential importance of other environmental controls, which
are already discussed at page 13.

Author response: We explored these questions and found that positive correlations between
annual NDVI_{max} and SWI were evident at 41.0 [39.5, 42.5] % of sites that greened, while
negative correlations were evident at 6.5 [5.0, 8.0] % of sites that browned. We added
corresponding text to the Results.

Figure 1 (b and c): It is not easy to identify the number of sample sizes with the continuous blue
colour palette. I suggest to use class breaks (e.g. [1-5] [5-10] [10-5] ... [25-30]) instead to
improve the visualization (and potentially a different colour).

Author response: We modified the color scheme in Figure 1 as suggested to more clearly depict
samples sizes. We similarly modified the color schemes in Figure 4 to use class breaks instead of
a continuous color palette.

Figure 3: I suggest to add a colour legend to the plot to make it is easier to read the figure. Please
clarify in the caption if the thresholds of $\alpha = 0.05$ and 0.10 for the dark and light shades
correspond to the significance level of the trends and correlations or to absolute trend changes
and correlations.

Author response: As requested, we added a color legend to the figure and clarified in the figure
caption that the dark and light shades correspond to the significance level associated with the
trends and correlations.

Matthias Forkel, Dresden, Germany, 6. January 2020

**Reviewer #3 (Remarks to the Author):**

Berner and colleagues present the results of a study aimed at using long-term and high resolution
satellite data to track changes in maximum 'greenness' (quantified in terms of the normalized

difference vegetation index) over the arctic. They find increased greening at 45% of the locations
studied, and correlate the greening with historic increases in air temperature. The results suggest
that the land surface of high-latitudes is greening, or at least ~50% of it is.

The manuscript is very well written, and the analysis clearly described. The strength of the
analysis is that the authors are working with Landsat data, which provides retrievals at very high
spatial resolution (30m). Previous studies focused on high latitude greening estimated from
remote sensing, of which there are many, used coarser resolution (~8km) data. Working with
such high resolution data is computationally challenging, as evidenced by the large amount of
code submitted with the manuscript. This is a very commendable effort and adds significant
value to the manuscript.

Author response: We thank Reviewer 3 for providing thoughtful feedback regarding our analysis
and manuscript, as well as for recognizing the value of our study and the myriad challenges
associated with conducting the study. Based on this feedback we made several important changes
to the analysis and manuscript, specifically:

(1) We revised our technique for estimating Landsat NDVI_{max} so that for each site we now fit
a series of decadal scale phenological curves rather than imposing a single mean seasonal
cycle derived from the full time series. This change had little impact on the resulting
NDVI_{max} time series, though we believe it is a more robust approach.

(2) We assessed the relationship between Landsat NDVI_{max} and both current and 2-year
average summer temperatures (SWI) using raw and detrended time series. These
assessments showed the interannual covariation between NDVI_{max} and SWI was strongly
influenced by the presence of positive trends in both time series, but that NDVI_{max} and
SWI still co-varied after removing long-term trends. Moreover, the assessments showed
NDVI_{max} was more strongly related to 2-year average SWI than current-year SWI, even
when detrended, thus highlighting multi-year effects of summer temperature on tundra
greenness.

(3) To add further novelty to our analysis, we explored climatic, topographic, permafrost,
and fire-related factors that contributed to spatial variability in NDVI trends across the
Arctic. This analysis highlighted that changes in summer temperature, annual ground
temperature, and summer minimum soil moisture were the most important examined
factors contributing to spatial patterns of greening and browning in the Arctic. Fires were
locally important, but so uncommon as to have little impact at a pan-Arctic scale.

We provide additional details below.

My main concerns regard some of the methods employed, and the novelty.

In particular, the spatial resolution of the Landsat data comes at the expense of temporal
coverage. As the authors note and develop methods to correct for, the sparse temporal coverage
also has a temporal dynamic. I.e. The further back you go in time the fewer retrievals there are.
With few retrievals it becomes difficult to estimate the maximum NDVI in each year. In order to

estimate maximum NDVI each year the authors therefore use a model informed by the mean
seasonal NDVI over all years. This could potentially introduce some bias, as the length of the
seasonal cycle has been extending over the decades due to warming. By imposing a mean
seasonal cycle (see Fig. S4), the authors are potentially overestimating the maximum NDVI in
the early parts of the timeseries, as the actual growing season was likely shorter than the
mean growing season that they impose. This could potentially lead to an underestimation of the
greening trend in the authors analysis. The authors could test this by imposing a temporal trend
in the mean seasonal cycle, through as far as I know the true change is unknown due to the
sparsity of retrievals. Note that the text frequently talks about pan-arctic but really much of
eastern Eurasia has no retrievals pre-2000, and western Eurasia has very spotty retrievals for
much of the prior decades also.

*Author response:* We fully agree with the reviewer that changes in the availability of Landsat
observations over time present a notable challenge to assessing NDVI trends across the Arctic
(or other regions). As the reviewer noted, this issue affects not only the domain over which we
can assess change through time, but also our ability to reliably estimate maximum summer
NDVI (NDVI_{max}) at individual sites even when observations are nominally available. We
attempted to address both issues in our study, but acknowledge the potential to improve aspects
of the analytical approach and refine aspects of the discussion. Consequently, we took several
steps to improve these aspects of the study during revision.

We developed a phenologically-informed approach to estimate NDVI_{max} at each site that
initially imposed a mean seasonal cycle of land surface phenology derived using all available
Landsat observations. As the reviewer noted, this approach could bias our estimates of NDVI_{max}
since the mean seasonal cycle was weighted towards more recent years given the greater
availability of observations. We therefore revised our approach to account for this potential
issue. Now rather than deriving a single mean seasonal cycle of land surface phenology for each
site, we instead use a moving-window approach to derive a mean seasonal cycle for every 17-
358 year interval from 1985 to 2016 (**Fig. S3**). We derived each mean seasonal cycle by fitting a
359 flexible cubic spline to the observations, which necessitated prescribing the flexibility of each
360 spline. To better account for uncertainty associated with this process, we randomly permuted
spline flexibility with each Monte Carlo simulation ($n = 10^3$). After revising our approach, we
then generated new time series of NDVI_{max} that we used to re-assess temporal trends and for
other aspects of the analysis.

 **Fig. S3** | Illustration of approach for estimating annual Landsat maximum summer NDVI
 ($NDVI_{max}$). (a) Seasonal progression of Landsat NDVI from June through August for a sampling
 site in the Arctic. Each point is a quality-controlled Landsat 5, 7, or 8 observation from 1985 to
 2016. Each curve depicts the typical land surface phenology for a 17-year period derived by
 fitting a cubic spline through all observations from that period. (b) Annual Landsat $NDVI_{max}$
 (black point) was estimated using each summer observation (brown points) together with
 phenological information on the typical difference in NDVI between peak summer and the
 timing of each observation [blue lines].

 These revisions and the introduction on sensor-specific calibration uncertainty led to a slight
 reduction in our estimate of the magnitude of change in mean Arctic $NDVI_{max}$ during recent
 decades, as well as impacted our estimates of the frequencies *greening* and especially *browning*
 across sites. For instance, we initially estimated that mean Arctic $NDVI_{max}$ increased 8.1% from
 1985 to 2016, while our revised analysis suggests an increase of 7.3 [7.0, 7.7] % during this
 period [95% Monte Carlo confidence interval]. Furthermore, our estimates of the percentage of
 sites that *greened* decreased from 45.0% to 37.3 [36.3, 38.4] %, while our estimates of the
 percentage of sites that *browned* increased from 3.3% to 4.7 [4.4, 5.2] %. We updated all trend
 estimates in our revised manuscript and modified the methods and supplemental material to
 describe our refined approach to estimating $NDVI_{max}$.

The reviewer also highlighted that Landsat observations were not widely available from
 parts of the Arctic before the year 2000, making it challenging to infer pan-Arctic trends in
 NDVI using Landsat. We assessed NDVI trends where possible from 1985 – 2016; however, the
 paucity of Landsat of across eastern and parts of western Eurasia pre-2000 also led us to assess
 trends from 2000 – 2016 when a fuller pan-Arctic assessment is more feasible. In revising the
 manuscript, we modified statements about pan-Arctic trends to better acknowledge the lack of
 pre-2000 observations for portions of the domain. This include the following statements:

[Introduction] It is important to note that Landsat observations were available for ~64%
 and ~96% of the Arctic domain from 1985 to 2016 and 2000 to 2016, respectively, with
 particularly improved coverage across the central and eastern Eurasian Arctic during the
 more recent period (**Error! Reference source not found.**b,c).

[Results] There was extensive greening in parts of western Eurasia (e.g. Yamal) and
 North America (e.g. Quebec) from 1985 to 2016, while the increase in availability of
 observations from 2000 to 2016 also revealed extensive greening in eastern Eurasia (e.g.,
 Yakutia; Fig. 4a,e).

[Discussion] Compared to AVHRR or MODIS, high resolution Landsat observations
 may be especially advantageous when assessing vegetation dynamics in highly
 heterogeneous tundra landscapes (Myers-Smith et al. 2020). Nevertheless, the paucity of
 Landsat observations in parts of the Arctic (e.g. eastern Eurasia) prior to 2000 constrain
 the utility of these sensors, as does the lower frequency of observations acquired each
 summer.

 The correlation with the SWI anomaly is interesting but potentially misleading. What the authors
 present as an anomaly is in fact a normalized change, in that the time series is not detrended. As
 can be seen from Fig. 2c there is no relationship between NDVI and SWI anomalies within
 decades (with the exception perhaps of the 2010s), it is only across decades that the relationship
 emerges due to the lack of detrending. What this analysis shows therefore is that both NDVI and
 SWI are trending, but it does not show that one is sensitive to the other (as claimed by the
 authors). In order to show that NDVI is sensitive to changes in SWI then you should detrend
 both NDVI and SWI. Otherwise, you should revise the text to say that both are trending up, and
 although there is no correlation between one and the other on an interannual (or multi-annual)
 scale there is reason to believe they are related over longer timescales.

Author response: The reviewer raises a valuable point that the co-variation between annual
 tundra greenness ($NDVI_{max}$) and summer temperature (SWI) can be affected by trends in both
 time series. We therefore re-evaluated correlations between $NDVI_{max}$ and SWI after linearly
 detrending each time series. Furthermore, to examine potential multi-year effects of summer
 temperatures and tundra greenness, we also examined the correlations between $NDVI_{max}$ and
 two-year average SWI using both raw and detrended time series. Moreover, we now quantify
 how NDVI and SWI uncertainty affects co-variation. This was accomplished by computing 10^3
 realizations of each correlation, where each realization involved randomly permuting NDVI and
 SWI time series. These substantive additions provide further insight into links between tundra
 greenness and summer temperatures across the Arctic.

Further analysis revealed several notable findings (Table S7). First, the $NDVI_{max} - SWI$
 correlations were still positive, but notable weaker after detrending both time series. For
 instance, the Spearman correlation (r_s) between mean Arctic $NDVI_{max}$ and SWI decreased from
 0.68 [0.66, 0.70] to 0.43 [0.41, 0.45] after detrending both time series [95% confidence interval].
 Similar patterns were evident among the bioclimatic zones. Second, the $NDVI_{max} - SWI$

correlations were consistently stronger when $NDVI_{max}$ was compared against 2-year average
 SWI instead of current-year SWI. For example, the mean Arctic $NDVI_{max}$ – SWI correlation
 increased from 0.68 [0.66, 0.70] to 0.86 [0.85, 0.88] or from 0.43 [0.41, 0.45] to 0.72 [0.69, 0.75]
 after detrending the time series. These patterns were again evident among bioclimatic zones.
 These findings illustrate that co-variation between annual $NDVI_{max}$ and SWI was strongly
 affected by positive trends in both time series over the last several decades. Nevertheless,
 $NDVI_{max}$ co-varied with current year SWI and particularly with two-year average SWI even after
 detrending every time series. We reference these findings in the Results section and include both
 a summary table and an 8-panel set of maps in the supplemental material.

 Table S7. Co-variation in annual mean tundra greenness (Landsat $NDVI_{max}$) and summer air
 temperatures (SWI) for the Arctic and each bioclimatic zone during recent decades. Spearman
 correlations (r_s) were used to assess co-variation between $NDVI_{max}$ and both current year and 2-
 444 year average SWI. Co-variation was also assessed after linearly detrending both $NDVI_{max}$ and
 445 SWI. Each correlation coefficient is accompanied by a 95% confidence interval derived from a
 446 Monte Carlo analysis that involved generating 10^3 realizations of every correlation.

Period	Domain	Spearman correlation (r_s) between $NDVI_{max}$ and ...			
		current year SWI		two-year average SWI	
		with trends	detrended	with trends	detrended
1985-2016	Arctic	0.68 [0.66,0.70]	0.43 [0.41,0.45]	0.86 [0.85,0.88]	0.72 [0.69,0.75]
	High Arctic	0.63 [0.58,0.67]	0.58 [0.53,0.62]	0.73 [0.69,0.77]	0.65 [0.58,0.72]
	Low Arctic	0.61 [0.58,0.63]	0.31 [0.27,0.34]	0.83 [0.81,0.84]	0.63 [0.61,0.65]
	Oro Arctic	0.69 [0.66,0.71]	0.40 [0.36,0.45]	0.77 [0.75,0.79]	0.43 [0.40,0.47]
2000-2016	Arctic	0.76 [0.73,0.78]	0.39 [0.33,0.46]	0.89 [0.88,0.91]	0.68 [0.65,0.70]
	High Arctic	0.75 [0.69,0.80]	0.78 [0.72,0.85]	0.70 [0.64,0.77]	0.70 [0.61,0.77]
	Low Arctic	0.65 [0.61,0.69]	0.38 [0.33,0.44]	0.84 [0.78,0.88]	0.51 [0.46,0.56]
	Oro Arctic	0.70 [0.68,0.72]	0.46 [0.40,0.51]	0.88 [0.85,0.92]	0.52 [0.50,0.57]

 My final concern is one of novelty. There have been many many studies calculating trends in
 NDVI from remote sensing data. The authors motivate the study by the fact that previous studies
 of greening focus on coarser resolution retrievals, and the 30m resolution used here is needed to
 accurately quantify trends. But I did not really see where the strength of this high resolution led
 to conclusions that were not obtainable with the coarser resolution products. In the end the
 authors aggregate their results to three very coarse zones. The fine resolution leads to spatial and
 temporal data sparsity, so it could equally be argued that the coarser resolution satellites are
 better as they cover larger areas and have more frequent data availability. Although the authors

did a great job of using the fine resolution to compare to field data, I was disappointed that the
 emphasis on fine resolution did not lead to any new conclusions, and was not used to help
 understand the large differences between previously published results using coarser resolution
 data. Those differences are discussed in detail in a recent paper by one of the authors, Myers-
 Smith, published in Nature Climate Change (DOI: 10.1038/s41558-019-0688-1, not cited).
 Showing how the fine resolution leads to more defensible conclusions would be a great addition
 to this manuscript.

*Author response: To increase the novelty our study and better capitalize on Landsat’s relatively*
 *high spatial resolution, we performed an additional analysis that explored the extent to which*
 *select climatic, permafrost, topographic, land cover and fire-related factors influenced spatial*
 *variability in NDVI_{max} trends across the Arctic. For each site, we classified the NDVI_{max} trend*
 *from 2000 – 2016 as browning, no trend, or greening. We then developed Random Forest models*
 *to predict the trend class at each site based on variables shown in table S8. For time series*
 *variables, we included both the change over time and the model-fit value during the first year of*
 *observation. Since sampling sites with browning were much less common than greening or no*
 *trend, we balanced the sample sizes so that each Random Forest model was fit using an equal*
 *number of sampling sites in each class. Furthermore, we trained each model using 2/3rd of data*
 *and evaluated model performance using the remaining 1/3rd of data that were withheld (Tables*
 *S9 and S10).*

**Table S8 | Summary of environmental data sets used with random forest models to predict**
 **Landsat NDVI_{max} trends from 2000 to 2016 at each sampling site.** These geospatial data sets
 span the pan-Arctic domain.

Theme	Variable	Units	Period	Cadence	Resolution
Climate	Summer warmth index	°C	2000-2016	Annual	50 km
	Minimum summer soil moisture	mm	2000-2016	Annual	4 km
Permafrost	Active layer thickness	cm	2003-2016	Annual	1 km
	Soil temperature at 1 m depth	°C	2003-2016	Annual	1 km
	Permafrost extent	%	2003-2016	Annual	1 km
	Thermokarst vulnerability	category	ca. 2015	Single time	--
Biological	Land cover	category	2015	Since time	0.3 km
Fire	Burned area	category	2001-2016	Monthly	0.5 km
Topography	Elevation	m	ca. 2015	Single time	0.09 km
	Slope	°	ca. 2015	Single time	0.09 km
	Aspect	°	ca. 2015	Single time	0.09 km
	Topographic Roughness Index	unitless	ca. 2015	Single time	0.09 km

Topographic Position Index	unitless	ca. 2015	Single time	0.09 km
----------	----------	-------------	---------

479 We added the following text, figures, and tables to the Results:

To further explore potential drivers of changes in tundra greenness among sampling sites,
we constructed Random Forest models to associate the NDVI_{max} trend category from
2000 to 2016 (i.e., browning, no trend, or greening) with climate, permafrost, land cover,
topography, and fire (Table S9). Cross-validated model classification accuracy was 55
[53, 58] %, but the classification accuracies for greening and browning classes were 70
[68, 73] % and 73 [70, 75] %, respectively (Table S10, S11). The expected classification
accuracy at random would be 33.3%. The six most important predictor variables included
change in SWI (2000-2016), soil temperature (1 m depth) and SWI in the early 2000s,
elevation, change in minimum summer soil moisture (2000-2016), and change in soil
temperature (2003-2016; **Fig. 4a**). Greening occurred more often at warm, high-elevation
sampling sites with increased air temperatures, soil temperatures, and summer soil
moisture. Conversely, browning occurred more often at cold, low-elevation sampling
sites with decreasing air temperature, soil temperature, and summer soil moisture (**Fig.**
**4b**). A notable exception was the sharp decline in greening and increase in browning
where soil temperatures in the early 2000s exceeded 0°C. It is also notable that at a pan-
Arctic scale recent fires were not an important predictor of greening or browning,
reflecting the fact that fires occurred at only ~1.1 % of sampling sites from 2001 to 2016.

**Table S9 | Class-specific performance of Random Forest models used to predict the**
 **Landsat NDVI_{max} trend from 2000 to 2016 at each sampling site.** The Landsat NDVI_{max} trend
 at each sampling site was classified as browning, no trend, or greening based on the direction and
 significance ($\alpha = 0.10$) of trend evaluated using a Mann-Kendal Trend Test and Theil-Sen Slope.
 A Random Forest model was fit for each of the 10^3 Monte Carlo simulations after balancing the
 number of sampling sites in each trend class. The performance of each model was assessed by
 withholding a random 33.3% of data for cross-validation. The overall cross-validated model
 classification accuracy was 55.4 [53.1, 57.5] %. The table summarizes the median and 95%
 confidence intervals for class-specific model sensitivity, specificity, and balanced accuracy
 derived from these simulations.

NDVI_{max} trend	Sensitivity	Specificity	Balanced Accuracy
Browning	65.6 [61.1, 69.9] %	79.6 [76.5, 82.3] %	72.6 [70.3, 74.5] %
No trend	38.4 [33.4, 43.2] %	75.5 [71.7, 79.0] %	56.9 [54.6, 59.0] %
Greening	62.3 [57.1, 66.9] %	78.1 [75.2, 80.9] %	70.2 [67.9, 72.5] %

**Table S10 | Confusion matrix comparing observed and predicted Landsat NDVI_{max} trends**
 **at sampling sites using predictions from Random Forest models.** The Landsat NDVI_{max} trend
 at each sampling site was classified as browning, no trend, or greening based on the direction and
 significance ($\alpha = 0.10$) of trend evaluated using Mann-Kendal Trend Test and Theil-Sen Slope.
 A confusion matrix was generated for each random forest model (10^3 Monte Carlo simulations)
 by withholding a random 33.3% of data for cross-validation. The table summarizes the median
 [95% CI] number of sampling sites falling in each category.

Observed	Predicted NDVI_{max} trend		
	Browning	No trend	Greening
NDVI_{max} trend			
Browning	431 [394, 465]	153 [121, 183]	74 [55, 92]
No trend	190 [160, 225]	252 [222, 281]	214 [178, 250]
Greening	77 [60, 100]	170 [135, 207]	410 [371, 449]

 **Fig. 4 | Variable importance and partial dependence of top six most important variables for**
 **classifying site-level Landsat NDVI_{max} trends from 2000 to 2016 using Random Forests.** The
 three NDVI_{max} trend categories included *browning*, *no trend*, and *greening* that were based on
 trend direction and significance ($\alpha = 0.10$). **(a)** Variable importance was characterized by the
 mean decrease in accuracy, where a higher value indicates that a variable is more important to
 the classification. **(b)** Partial dependency plots illustrate how each predictor variable affects class
 probability while accounting for the mean effect of other predictors in the model. The top six
 predictor variables included changes (Δ) in summer warmth index (2000-2016), summer soil
 moisture (2000-2016), and annual mean soil temperature (2003-2016), as well as elevation,
 summer warmth index in 2000, and annual mean soil temperature in 2003. Soil temperature data
 were for 1 m depth and were not available prior to 2003. Boxplot lines and whiskers in **(a)**
 depict median estimates and 95% confidence intervals derived from Monte Carlo simulations ($n = 10^3$),
 as do solid lines and error bands in **(b)**.

 We added a corresponding Methods section and further discussion of potential drivers in the
 Discussion, including a new paragraph about the role of fires. An important current limitation to
 assessing drivers of greening and browning is the lack of pan-Arctic environmental data sets at
 high spatial and or thematic resolution. We now include a call in the Discussion for additional
 efforts along these lines:

Attribution of local tundra greening or browning to specific biological or environmental
 drivers remains an important challenge that will require further mapping and modeling of
 potential drivers at higher spatial and thematic resolution, coupled with field
 measurements, across the Arctic.

 In addition to this analysis, we now do a better job of highlighting instance where our
 comparisons between satellite and field measurements provide additional support for changes
 observed by the satellites. For instance, in examining difference among satellites, we note:

... Our Landsat analysis also indicated recent browning along the southwestern coast of
 Greenland that is consistent with local declines in shrub growth(Gamm et al. 2018), but
 not evident in assessments that used AVHRR or MODIS. The link between recent
 browning and declining shrub growth is further supported by the positive correlations that
 we found between annual Landsat NDVI_{max} and stem radial growth of grayleaf willow
 (*Salix glauca*; $r_s = 0.60$ [0.39, 0.78]) and dwarf birch (*Betula nana*; $r_s = 0.61$ [0.45, 0.74])
 in this region (Table S2).

Also, while discussing possible impacts of greening on wildlife, we highlight:

Our Landsat analysis showed tundra greening in regions with potential moose, beaver,
 caribou, and reindeer habitat and demonstrated that variability in tundra greenness was
 often associated with annual shrub growth in these regions (Table S5).

We now cite Myers-Smith et al. (2020) in the several parts of the Discussion, including in the
 following locations:

- • While our Landsat analysis revealed an increase in mean Arctic NDVI_{max} that broadly
 supports macrosystem changes inferred using coarser-resolution AVHRR (1982 onward
 at ~8 km resolution) and MODIS (2000 onward at 500 m resolution) data sets, it also
 highlights several important inconsistencies among satellites (Guay et al. 2014, Bhatt et
 al. 2017, Myers-Smith et al. 2020).
- • Compared to AVHRR or MODIS, high resolution Landsat observations may be
 especially advantageous when assessing vegetation dynamics in highly heterogeneous
 tundra landscapes(Myers-Smith et al. 2020).
- • Lastly, tundra greenness could, in some areas, be confounded by patchy vegetation
 interspersed with bare ground, surface water, or snow(Raynolds and Walker 2016,
 Myers-Smith et al. 2020). Despite limitations with NDVI (e.g., see (Myers-Smith et al.
 2020) for recent tundra-specific review), our results indicate Arctic plants did not
 universally benefit from warming in recent decades, highlighting variable plant
 community responses to warming mediated by a combination of biotic and abiotic
 factors.

Minor comments:

For future submissions note that line numbers are always appreciated.

Author response: We apologize for accidentally overlooking line numbers and rectified this
 oversight in the revised manuscript!

Page 2: “resolve disparate finding(s)”

Author response: We made “findings” plural.

Page 2: The references to the field studies are somewhat out of date. See for example: DOI:
10.1002/ecm.1351

Author response: We removed three references published 2010-2015 and added three references
to field studies published 2017-2019, including Forchhammer (2017), Myers-Smith et al. (2019)
and Bjorkman et al. (2018).

Page 7: The correlation between NDVImax and SWI anomalies mostly seems to be driven by the
increasingly positive anomalies in SWI over time. Ie. As NDVI trends then it's not surprising
that there is a correlation with another trending variable. But should the SWI anomalies be
trending?

Author response: Please see our response to your previous question on this matter.

Literature cited

- Beck, P. S. A., and S. J. Goetz. 2011. Satellite observations of high northern latitude vegetation
productivity changes between 1982 and 2008: ecological variability and regional differences.
*Environmental Research Letters* **6**:045501.
- Berner, L. T., P. S. A. Beck, A. G. Bunn, and S. J. Goetz. 2013. Plant response to climate change along the
forest-tundra ecotone in northeastern Siberia. *Global Change Biology* **19**:3449-3462.
- Berner, L. T., P. S. A. Beck, A. G. Bunn, A. H. Lloyd, and S. J. Goetz. 2011. High-latitude tree growth and
satellite vegetation indices: Correlations and trends in Russia and Canada (1982–2008). *Journal*
*of Geophysical Research* **116**:G01015.
- Bhatt, U. S., D. A. Walker, M. Reynolds, P. Bieniek, H. E. Epstein, J. Comiso, J. Pinzon, C. Tucker, M.
Steele, and W. Ermold. 2017. Changing seasonality of panarctic tundra vegetation in relationship
to climatic variables. *Environmental Research Letters* **12**:1-18.
- Bjorkman, A. D., I. H. Myers-Smith, S. C. Elmendorf, S. Normand, N. Rüger, P. S. A. Beck, A. Blach-
Overgaard, D. Blok, J. H. C. Cornelissen, B. C. Forbes, D. Georges, S. J. Goetz, K. C. Guay, G. H. R.
Henry, J. HilleRisLambers, R. D. Hollister, D. N. Karger, J. Kattge, P. Manning, J. S. Prevéy, C.
Rixen, G. Schaepman-Strub, H. J. D. Thomas, M. Vellend, M. Wilmking, S. Wipf, M. Carbognani, L.
Hermanutz, E. Lévesque, U. Molau, A. Petraglia, N. A. Soudzilovskaia, M. J. Spasojevic, M.
Tomaselli, T. Vowles, J. M. Alatalo, H. D. Alexander, A. Anadon-Rosell, S. Angers-Blondin, M. t.
Beest, L. Berner, R. G. Björk, A. Buchwal, A. Buras, K. Christie, E. J. Cooper, S. Dullinger, B.
Elberling, A. Eskelinen, E. R. Frei, O. Grau, P. Grogan, M. Hallinger, K. A. Harper, M. M. P. D.
Heijmans, J. Hudson, K. Hülber, M. Iturrate-Garcia, C. M. Iversen, F. Jaroszynska, J. F. Johnstone,
R. H. Jørgensen, E. Kaarlejärvi, R. Klady, S. Kuleza, A. Kulonen, L. J. Lamarque, T. Lantz, C. J. Little,
618 J. D. M. Speed, A. Michelsen, A. Milbau, J. Nabe-Nielsen, S. S. Nielsen, J. M. Ninot, S. F.
Oberbauer, J. Olofsson, V. G. Onipchenko, S. B. Rumpf, P. Semenchuk, R. Shetti, L. S. Collier, L. E.
Street, K. N. Suding, K. D. Tape, A. Trant, U. A. Treier, J.-P. Tremblay, M. Tremblay, S. Venn, S.
Weijers, T. Zamin, N. Boulanger-Lapointe, W. A. Gould, D. S. Hik, A. Hofgaard, I. S. Jónsdóttir, J.
Jorgenson, J. Klein, B. Magnusson, C. Tweedie, P. A. Wookey, M. Bahn, B. Blonder, P. M. van
Bodegom, B. Bond-Lamberty, G. Campetella, B. E. L. Cerabolini, F. S. Chapin, W. K. Cornwell, J.
Craine, M. Dainese, F. T. de Vries, S. Díaz, B. J. Enquist, W. Green, R. Milla, Ü. Niinemets, Y.
Onoda, J. C. Ordoñez, W. A. Ozinga, J. Penuelas, H. Poorter, P. Poschlod, P. B. Reich, B. Sandel, B.

- Schamp, S. Sheremetev, and E. Weiher. 2018. Plant functional trait change across a warming
tundra biome. *Nature* **562**:57-62.
- Bunn, A. G., and S. J. Goetz. 2006. Trends in Satellite-Observed Circumpolar Photosynthetic Activity from
1982 to 2003: The Influence of Seasonality, Cover Type, and Vegetation Density. *Earth*
*Interactions* **10**:1-19.
- Bunn, A. G., S. J. Goetz, and G. J. Fiske. 2005. Observed and predicted responses of plant growth to
climate across Canada. *Geophysical Research Letters* **32**:L16710.
- Forchhammer, M. 2017. Sea-ice induced growth decline in Arctic shrubs. *Biology Letters* **13**.
- Gamm, C. M., P. F. Sullivan, A. Buchwal, R. J. Dial, A. B. Young, D. A. Watts, S. M. Cahoon, J. M. Welker,
and E. Post. 2018. Declining growth of deciduous shrubs in the warming climate of continental
western Greenland. *Journal of Ecology* **106**:640-654.
- Goetz, S. J., A. G. Bunn, G. J. Fiske, and R. A. Houghton. 2005. Satellite-observed photosynthetic trends
across boreal North America associated with climate and fire disturbance. *Proceedings of the*
*National Academy of Sciences of the United States of America* **102**:13521-13525.
- Guay, K. C., P. S. A. Beck, L. T. Berner, S. J. Goetz, A. Baccini, and W. Buermann. 2014. Vegetation
productivity patterns at high northern latitudes: a multi-sensor satellite data assessment. *Global*
*Change Biology* **20**:3147-3158.
- Ju, J., and J. G. Masek. 2016. The vegetation greenness trend in Canada and US Alaska from 1984–2012
Landsat data. *Remote Sensing of Environment* **176**:1-16.
- Karlsen, S. R., H. B. Anderson, R. Van der Wal, and B. B. Hansen. 2018. A new NDVI measure that
overcomes data sparsity in cloud-covered regions predicts annual variation in ground-based
estimates of high arctic plant productivity. *Environmental Research Letters* **13**:025011.
- Myers-Smith, I. H., M. M. Grabowski, H. J. D. Thomas, S. Angers-Blondin, G. N. Daskalova, A. D.
Bjorkman, A. M. Cunliffe, J. J. Assmann, J. S. Boyle, E. McLeod, S. McLeod, R. Joe, P. Lennie, D.
Arey, R. R. Gordon, and C. D. Eckert. 2019. Eighteen years of ecological monitoring reveals
multiple lines of evidence for tundra vegetation change. *Ecological Monographs* **89**:e01351.
- Myers-Smith, I. H., J. T. Kerby, G. K. Phoenix, J. W. Bjerke, H. E. Epstein, J. J. Assmann, C. John, L. Andreu-
Hayles, S. Angers-Blondin, P. S. A. Beck, L. T. Berner, U. S. Bhatt, A. D. Bjorkman, D. Blok, A. Bryn,
C. T. Christiansen, J. H. C. Cornelissen, A. M. Cunliffe, S. C. Elmendorf, B. C. Forbes, S. J. Goetz, R.
D. Hollister, R. de Jong, M. M. Loranty, M. Macias-Fauria, K. Maseyk, S. Normand, J. Olofsson, T.
C. Parker, F.-J. W. Parmentier, E. Post, G. Schaepman-Strub, F. Stordal, P. F. Sullivan, H. J. D.
Thomas, H. Tømmervik, R. Treharne, C. E. Tweedie, D. A. Walker, M. Wilmking, and S. Wipf.
2020. Complexity revealed in the greening of the Arctic. *Nature Climate Change* **10**:106-117.
- Myneni, R. B., C. D. Keeling, C. J. Tucker, G. Asrar, and R. R. Nemani. 1997. Increased plant growth in the
northern high latitudes from 1981 to 1991. *Nature* **386**:698-702.
- National Academies of Sciences. 2019. Understanding Northern Latitude Vegetation Greening and
Browning: Proceedings of a Workshop. The National Academies Press, Washington, DC.
- Nitze, I., and G. Grosse. 2016. Detection of landscape dynamics in the Arctic Lena Delta with temporally
dense Landsat time-series stacks. *Remote Sensing of Environment* **181**:27-41.
- Pattison, R. R., J. C. Jorgenson, M. K. Reynolds, and J. M. Welker. 2015. Trends in NDVI and Tundra
Community Composition in the Arctic of NE Alaska Between 1984 and 2009. *Ecosystems* **18**:707-
719.
- Pinzon, J., and C. Tucker. 2014. A Non-Stationary 1981–2012 AVHRR NDVI3g Time Series. *Remote*
*Sensing* **6**:6929-6960.
- Reynolds, M. K., and D. A. Walker. 2016. Increased wetness confounds Landsat-derived NDVI trends in
the central Alaska North Slope region, 1985–2011. *Environmental Research Letters* **11**:085004.

Roy, D. P., V. Kovalskyy, H. K. Zhang, E. F. Vermote, L. Yan, S. S. Kumar, and A. Egorov. 2016.
Characterization of Landsat-7 to Landsat-8 reflective wavelength and normalized difference
vegetation index continuity. *Remote Sensing of Environment* **185**:57-70.

REVIEWERS' COMMENTS:

Reviewer #3 (Remarks to the Author):

Thank you to the authors for their detailed consideration of my suggestions and concerns. The responses and changes made to the manuscript in response to all comments have greatly improved it, particularly the additional text highlighting the utility of Landsat compared to AVHRR, and the additional analysis to improve the robustness of the results. I am happy to say that I have no further comments and congratulate the authors on an excellent study.

T. Keenan (reviewer #3)

Responses to reviewer comments

REVIEWERS' COMMENTS:

Reviewer #3 (Remarks to the Author):

Thank you to the authors for their detailed consideration of my suggestions and concerns. The responses and changes made to the manuscript in response to all comments have greatly improved it, particularly the additional text highlighting the utility of Landsat compared to AVHRR, and the additional analysis to improve the robustness of the results. I am happy to say that I have no further comments and congratulate the authors on an excellent study. T. Keenan (reviewer #3)

- Author response: We appreciate the constructive feedback provided by Dr. Keenan and similarly believe the revisions greatly improved the analysis and paper.